# The liver microenvironment orchestrates FGL1-mediated immune escape and progression of metastatic colorectal cancer

Jia-Jun Li [1,5], Jin-Hong Wang[1,5], Tian Tian[2,5], Jia Liu[1,5], Yong-Qiang Zheng [1], Hai-Yu Mo[1], Hui Sheng[1], Yan-Xing Chen[1], Qi-Nian Wu[1], Yi Han[3], Kun Liao[1], Yi-Qian Pan[1], Zhao-Lei Zeng [1], Ze-Xian Liu [1], Wei Yang[3], Rui-Hua Xu [1,4] ✉ & Huai-Qiang Ju [1,4] ✉

Colorectal cancer (CRC) patients with liver metastases usually obtain less benefit from immunotherapy, and the underlying mechanisms remain understudied. Here, we identify that fibrinogen-like protein 1 (FGL1), secreted from cancer cells and hepatocytes, facilitates the progression of CRC in an intraportal injection model by reducing the infiltration of T cells. Mechanistically, tumor-associated macrophages (TAMs) activate NF-κB by secreting TNFα/IL-1β in the liver microenvironment and transcriptionally upregulate OTU deubiquitinase 1 (OTUD1) expression, which enhances FGL1 stability via deubiquitination. Disrupting the TAM-OTUD1-FGL1 axis inhibits metastatic tumor progression and synergizes with immune checkpoint blockade (ICB) therapy. Clinically, high plasma FGL1 levels predict poor outcomes and reduced ICB therapy benefits. Benzethonium chloride, an FDA-approved antiseptics, curbs FGL1 secretion, thereby inhibiting liver metastatic tumor growth. Overall, this study uncovers the critical roles and posttranslational regulatory mechanism of FGL1 in promoting metastatic tumor progression, highlighting the TAM-OTUD1-FGL1 axis as a potential target for cancer immunotherapy.

Colorectal cancer (CRC) is the third most commonly diagnosed cancer and second leading cause of cancer mortality globally[1,2]. During the disease course of colorectal cancer patients, approximately 50% develop liver metastases, which remains the major cause of death from cancer and heralds a poor prognosis with a 5-year survival rate of less than 20%[1,3]. Cancer immunotherapy has transformed the clinical landscape for the treatment of several cancer types and induces durable antitumor responses in subsets of cancer patients[4]. Individuals with colorectal cancer liver metastasis (CRLM) are more likely to exhibit a suppressive immune microenvironment and experience less clinical benefit from immunotherapy[5]. Additionally, understanding the organ-specific tumor immune contexture has been one of the top ten key challenges for cancer immunotherapy[6]. The liver harbors a unique architecture of immune tolerance, and the tumor microenvironment (TME) within liver metastasis reduces anticancer immunity[5,7]. However, their roles in CRLM and the underlying mechanisms need to be further explored.

Immune checkpoint blockade (ICB) can disrupt the immune escape of cancer cells by removing inhibitory signals of T-cell activation and improve patient survival in many cancer types[8]. Cytotoxic T-lymphocyte antigen 4 (CTLA-4) and programmed cell death protein 1 (PD-1) are inhibitory molecules that attenuate T-cell activation via

[1]State Key Laboratory of Oncology in South China, Sun Yat-sen University Cancer Center, Guangzhou 510060, P. R. China. [2]Department of Medical Biochemistry and Molecular Biology, School of Medicine, Jinan University, Guangzhou, 510632 Guangdong, China. [3]Research Department of Medical Sciences, Guangdong Provincial People's Hospital, Guangdong Academy of Medical Sciences, Guangzhou, 510080 Guangdong, China. [4]Research Unit of Precision Diagnosis and Treatment for Gastrointestinal Cancer, Chinese Academy of Medical Sciences, Guangzhou, 510060 Guangdong, China. [5]These authors contributed equally: Jia-Jun Li, Jin-Hong Wang, Tian Tian, Jia Liu. ✉e-mail: xurh@sysucc.org.cn; juhq@sysucc.org.cn

different mechanisms[9]. mAbs targeting CTLA-4 and/or PD-1/PD-L1 have demonstrated promise in a variety of malignancies, including microsatellite instability-high (MSI-H) CRC, which comprises only 5% of metastatic CRC[1]. In addition, several novel immune checkpoint molecules have been identified, including lymphocyte activating 3 (LAG-3), T-cell immunoglobulin domain and mucin domain-3 (TIM3), V-domain Ig suppressor of T-cell activation (VISTA), and T-cell immunoglobulin and ITIM domain (TIGIT)[9,10]. Evaluating these targets is an attractive possibility for cancer treatment, and their therapeutic potential is being extensively investigated preclinically and clinically. Among these immunosuppressive molecules, identification of those that function as key molecules driving the immune evasion and progression of metastatic tumors in the liver microenvironment remains elusive.

Fibrinogen-like protein 1 (FGL1), also named hepatocyte-derived fibrinogen-like protein 1 (HFREP1), was originally found in hepatocytes and mainly contributes to the mitogenic and metabolic activity of hepatocytes under normal physiological conditions[11,12]. In recent years, high FGL1 expression was also found in tumor cells. FGL1 may simultaneously serve as an oncogene and tumor suppressor gene in various cancers due to tumor heterogeneity[13,14]. High FGL1 expression is associated with gefitinib resistance in non-small cell lung cancer (NSCLC), whereas low FGL1 expression is associated with sorafenib resistance in hepatocellular carcinoma (HCC), suggesting that FGL1 also plays significant roles in tumor therapy resistance[11]. Importantly, a recent study proved that FGL1 is a conserved and specific ligand of LAG-3, and its blockade can potentiate antitumor T-cell responses[15], emphasizing the potential of targeting FGL1/LAG-3 as the next generation of ICB therapy. Mechanistically, FGL1 has been reported to be transcriptionally regulated by IL-6-mediated JAK2/STAT3 signaling[11,14]. Nevertheless, the potential role of FGL1 in the immune escape of metastatic tumors in the liver microenvironment has not been investigated, and its regulation remains unclear.

In this study, we show that FGL1 facilitates the progression of CRC cells by reducing anticancer immunity in the liver microenvironment. Mechanistically, we reveal that tumor-associated macrophages (TAMs) mediate the stabilization of FGL1 through nuclear factor kappa-B (NF-κB) activation and OTU deubiquitinase 1 (OTUD1) expression in the liver microenvironment. Importantly, we prove the antitumor effects of the FDA-approved drug benzethonium chloride, highlighting the TAM-OTUD1-FGL1 axis as a potential target for liver metastatic cancer immunotherapy.

## Results

### FGL1 facilitates the progression of colorectal cancer cells through immunosuppression in the liver microenvironment

To identify key immunosuppressive molecules that promote CRC progression in the liver microenvironment, we first constructed a liver metastasis mouse model of murine CRC by microsurgical orthotopic implantation of MC38-GFP cells into the cecum termini of C57BL/6 J mice. As shown in Fig. 1A, we separately isolated MC38 cells from primary and liver metastatic tumor tissue or hepatocytes from control and metastatic liver tissue by a two-step collagenase perfusion technique and fluorescence-activated cell sorting (FACS) assay as reported[16]. By performing pathological and immunoblotting (IB) analyses, we detected the protein levels of seven major reported immunosuppressive molecules in the primary and liver metastatic MC38 tumor cells, including FGL1, VISTA, programmed death-ligand 1/2 (PD-L1/2), high mobility group Box 1 (HMGB1), nectin cell adhesion molecule 2 (Nectin2) and galectin-9 (Gal-9)[10]. The results show that FGL1 protein expression was distinctly elevated in liver metastatic tumor cells (Fig. 1B, C, S1A), but not elevated in hepatocytes from metastatic livers (Fig. S1B), which secrete FGL1 under normal physiological conditions[15]. This result was verified by higher plasma FGL1 levels in mice with liver metastases, as detected by ELISA (Fig. S1C). Similar results were found in BALB/c mice orthotopically implanted with CT26

cells (Fig. S1A–C). These observations indicate that FGL1 plays a potential role in facilitating the progression of CRC in the liver microenvironment.

A previous study reported that FGL1 inhibits antigen-specific T-cell activation and serves as an immunosuppressive molecule for the regulation of immune homeostasis[15]; thus, we speculated that elevated FGL1 might mediate metastatic CRC progression through immunosuppression. Owing to the low incidence of liver metastasis in orthotopic mouse model of CRC[17,18], we constructed the intraportal transplantation model to evaluate the effects of FGL1 on tumor burden using immunocompetent syngeneic mice (C57BL/6 J) and immunodeficient Rag1[-/-] mice, which lack mature B and T cells due to V(D)J recombination deficiency[19]. The results show that loss of FGL1 in MC38 cells significantly suppressed metastatic tumor progression in C57BL/6 J mice, as evidenced by the bioluminescence intensity, liver weight and number of liver metastases (Fig. 1D–H). In contrast, the inhibitory effect was not observed in Rag1[-/-] mice in vivo or in MC38 cells in vitro (Fig. S1D, E), suggesting that FGL1 plays a protumor role in the liver microenvironment through adaptive immunity. Notably, the T-cell-mediated killing of MC38-OVA cells was significantly enhanced by knockdown of FGL1 (Fig. S1F). Given that hepatocytes have been reported to secrete FGL1 in the liver microenvironment[15], we knocked down FGL1 in hepatocytes via injecting recombinant adeno-associated virus (AAV) targeting hepatocyte-specific FGL1 and found that metastatic tumor progression was also significantly suppressed (Fig. 1E–H, S1G). The observed weaker effects on the reduction of liver metastases upon downregulation of FGL1 in hepatocytes may be attributed to the time required to achieve effective knockdown[20]. Consistently, the plasma FGL1 levels were decreased in the mice with FGL1 knockdown in MC38 cells and hepatocytes (Fig. 1I). Considering that Rag1[-/-] mice have normal functioning dendritic cells (DCs), macrophages, and natural killer (NK) cells and lack T cells[21], we hypothesized that the difference in metastatic tumor growth inhibition was caused by T cells. We next analyzed tumor-infiltrating lymphocytes via flow cytometric analysis (Fig. S1H) and found that FGL1 depletion enhanced the infiltration of IFN-γ[+] CD8[+]/CD4[+] and Ki67[+] CD8[+]/CD4[+] T cells within the tumor microenvironment (Fig. 1J, S1I). Furthermore, FGL1 binds to LAG-3 in vitro (Fig. S1J). Notably, LAG-3 can be expressed by exhausted liver-recruited T cells (Fig. S1K), and inclusion of recombinant murine FGL1 protein (mFGL1) significantly reduced IFN-γ production by activated splenic T cells, which express high levels of LAG-3, and blockade of LAG-3 greatly rescued the reduced levels of IFN-γ production (Fig. S1L), indicating that FGL1 impedes the antitumor T cell response by binding to LAG-3 in liver metastases. In addition, knockdown of FGL1 both in MC38 cells and hepatocytes, which are the main sources of hepatic FGL1 in liver metastases, resulted in a significant prolongation of overall survival (OS) in mice compared to the control groups (Fig. 1K). Taken together, these data suggest that FGL1 promotes metastatic tumor growth by reducing tumor-infiltrating T cells in the liver microenvironment.

### High FGL1 levels predict poor outcomes and less benefit from PD-1/PD-L1 blockade therapy

We further investigated the clinical relevance of FGL1 protein or plasma levels in gastrointestinal tumors, including CRC, GC and esophageal squamous cell carcinoma (ESCC). The results show that FGL1 expression levels were upregulated in tumor tissue and were also significantly increased in liver metastatic tissues compared with paired primary tissues in CRC and GC, analyzed using a tissue microarray (SYSUCC) (Fig. 2A–D). Consistently, the plasma FGL1 levels were significantly higher in cancer patients than in healthy donors, and the levels were also significantly increased in metastatic CRC or GC patients compared with patients without liver metastases (Fig. 2E, F, S2A). Strikingly, Kaplan–Meier survival analysis indicated that higher plasma FGL1 levels were associated with poor outcomes

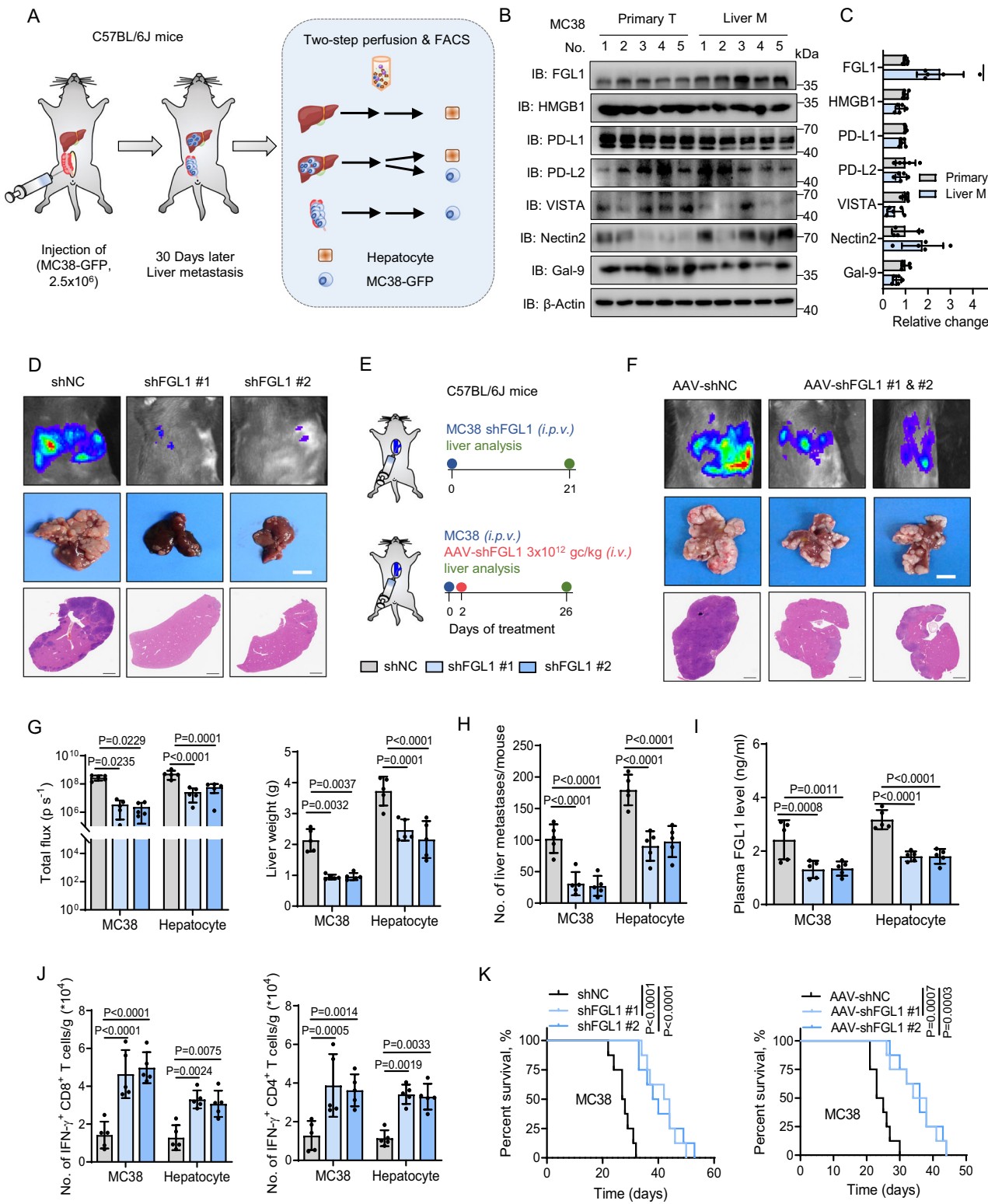

in CRC, GC and ESCC patients (Fig. 2G, S2B). Additionally, higher plasma FGL1 levels were correlated with diminished OS in CRC or GC patients with liver metastasis (Fig. 2H). In addition, we tested plasma FGL1 levels in four SYSUCC cohorts that received either anti-PD-1 plus chemotherapy or anti-PD-1 monotherapy to investigate whether FGL1 influences immunotherapy efficacy. In the neoadjuvant anti-PD-1 plus chemotherapy GC cohort, higher plasma FGL1 levels indicated a worse clinical stage of the tumor (Fig. 2I). In the first-line anti-PD-1 plus chemotherapy GC cohort, the first-line anti-PD-1 plus

chemotherapy advanced or metastatic ESCC cohort and the postline anti-PD-1 monotherapy advanced NPC cohort, higher plasma FGL1 levels were negatively correlated with the anti-PD-1 blockade therapy response (Fig. 2I, J). To confirm these findings, we further tested the association between the expression of FGL1 and the efficacy of immunotherapy in four published anti-PD-1/PD-L1 treatment cohorts[22–25]. We noted that FGL1 expression was inversely correlated with overall survival (Fig. 2K, L). Taken together, these results suggest that higher plasma FGL1 levels are associated with poor

**Fig. 1 | FGL1 facilitates the progression of colorectal cancer through immunosuppression in the liver microenvironment. A** Schematic of establishment of an orthotopic liver metastasis mouse model of CRC by injection of MC38-GFP cells into the cecum termini and the FACS sorting strategy. FACS, fluorescence-activated cell sorting. **B** Immunoblotting (IB) detection of indicated immunosuppressive molecule expression in MC38 cells sorted from primary and liver metastatic tumor tissue by flow cytometry. **C** Quantitative estimates of the expression of indicated immunosuppressive molecules using ImageJ based on IB analysis. **D** Representative luciferase images and H&E staining of metastatic livers from C57BL/6 J mice intraportally injected with the control (shNC) and FGL1 knockdown MC38 cells. Scale bar for bright-field images, 1 cm; scale bar for H&E images, 2.5 mm. **E** Experimental strategies. *i.p.v.*, intraportal vein injection; *i.v.*, intravenous injection. **F** Representative luciferase images and H&E staining of metastatic livers from C57BL/6 J mice intraportally implanted with MC38 cells and subjected to AAV-shNC or AAV-shFGL1 (#1, #2) treatment. Scale bar for bright-field images, 1 cm; scale bar

for H&E images, 2.5 mm. AAV, adeno-associated virus. **G** Bioluminescent quantification and liver weight of mice with knockdown of FGL1 in MC38 cells or hepatocytes (*n* = 5 mice per group). **H, I** Quantification of liver metastases (**H**) and ELISA detection of plasma FGL1 levels (**I**) in mice with FGL1 knockdown in MC38 cells and hepatocytes (*n* = 5 mice per group). **J** Flow cytometric analysis of the number of IFN-γ⁺ CD8⁺ and IFN-γ⁺ CD4⁺ T cells in liver metastases from mice with knockdown of FGL1 in MC38 cells or hepatocytes (*n* = 5 mice per group). **K** Survival curve analysis of mice in indicated groups (Mean survival times of shNC, shFGL1 #1, shFGL1 #2, AAV-shNC, AAV-shFGL1 #1 and AAV-shFGL1 #2 mice were 27.5, 41.6, 40.8, 24.5, 35 and 35.8 days, respectively) (*n* = 8 mice per group). IB experiments in B were repeated three times, the data are representative of three biologically independent experiments. The data in (**C, G–J**) are presented as the mean ± SD (*n* = 5). The data in K were determined by Kaplan–Meier analysis with the log-rank test. *P* values were determined by one-way ANOVA (**C, G–J**).

---

prognosis and may predict less clinical benefit from PD-1/PD-L1 blockade therapy, especially in patients with liver metastases.

## TAMs promote the stabilization of FGL1 by activating NF-κB in the liver microenvironment

We speculate that the abnormal upregulation of FGL1 in liver metastatic tumors is closely related to the liver microenvironment. Thus, we analyzed a previously published single-cell transcriptomic atlas of CRC liver metastases (LM)[26] and observed a markedly remodeled tumor immune microenvironment in LM compared with primary tumor tissue (PT) or primary normal tissue (PN) (Fig. 3A, B). Notably, quantitative analysis showed that macrophages were significantly enriched in LM (Fig. 3C). TAMs are known to promote tumor progression by suppressing antitumor immunity[27,28]. To determine whether TAMs are involved in FGL1 overexpression in liver metastatic tumors, we isolated TAMs from mouse or patient LM tissues and cocultured them indirectly with cancer cells (Fig. 3D). qPCR and flow cytometric analyses confirmed that TAMs from liver metastases show pro-tumorigenic (M2) subtype evidenced by elevated expression of *CD163*, *ARG-1* and *CD206* (Fig. S3A, B)[29]. IB analysis revealed that coculture with TAMs notably increased the FGL1 protein levels in CRC cells (Fig. 3D). In contrast, *FGL1* mRNA expression levels were not affected, which was also confirmed in liver metastatic tissues (Fig. 3E, F), indicating that TAM-secreted cytokines upregulate FGL1 protein expression via posttranslational regulation. After treatment with different cytokines routinely secreted by TAMs[27,28], we found that both TNFα and IL-1β induced FGL1 upregulation, which could be rescued by adding neutralizing antibodies to MC38 cells incubated with conditioned media produced by TAMs (TAM-CM) (Fig. 3G, S3C). Taken together, these data indicate that TAM-secreted TNFα and IL-1β upregulate FGL1 protein expression via posttranslational regulation in the liver microenvironment.

We next sought to determine the underlying mechanisms by which TAM mediates the upregulation of FGL1. First, we found that incubation with TAM-CM decreased FGL1 ubiquitination and prolonged its protein half-life in both MC38 and HT29 cells[30] (Fig. 3H, S3D). Because ubiquitin is a common denominator in the targeting of substrates to the two major protein degradation pathways in eukaryotic cells: proteasome and autophagy–lysosome[31], we explored which pathway was involved by using the lysosome inhibitor bafilomycin A1 (Baf-A1) and proteasome inhibitor MG132[29]. We observed that FGL1 degradation could be restored by Baf-A1 in MC38 cells treated with the protein synthesis inhibitor cycloheximide (CHX)[29], indicating that FGL1 protein stability was mainly controlled by the autophagy–lysosome pathway (Fig. S3E). Consistently, the addition of Earle's balanced salt solution (EBSS), which initiates the autophagy–lysosome pathway[32], remarkably accelerated colocalization puncta of FGL1-LC3B and degradation of FGL1 in cancer cells (Fig. S3F, G). TNFα/IL-1β activates several signaling pathways, including NF-κB, AKT, and mTOR[33,34], and

thus we applied several inhibitors to investigate which signaling cascade is involved in TAM-mediated FGL1 stabilization. The results prove that only BAY 11–7082, which inhibits IκB kinase β (IKKβ), abolished TAM-mediated FGL1 stabilization, suggesting that NF-κB may be involved in TAM-mediated FGL1 stabilization (Fig. 3I, S3H). Strikingly, we found that knockdown of p65 attenuated TAM-mediated FGL1 stabilization in accordance with depressed NF-κB activity, as indicated by decreased phosphorylated (P)-p65 levels (Fig. 3J). Similarly, TAMs failed to induce FGL1 stabilization in *p65⁻/⁻* and *IKK⁻/⁻* MEFs (Fig. 3K, S3I). In addition, knockdown of p65 increased FGL1 ubiquitination and shortened its protein half-life in cancer cells cultured with TAM-CM (Fig. 3L, M). Taken together, these results demonstrate that TAMs decrease FGL1 ubiquitination and enhance its stability via activation of the NF-κB/p65 signaling pathway in the liver microenvironment.

## Deubiquitinase OTUD1 is involved in TAM-mediated stabilization of FGL1

To identify the regulatory factor controlling TAM-mediated deubiquitination and stabilization of FGL1, we performed a coimmunoprecipitation (Co-IP) assay for TAM-cocultured cancer cells followed by mass spectrometry (MS). Among the FGL1-interacting proteins, the deubiquitinase OTUD1 was identified and verified by co-IP in 293T and MEF cells (Fig. 4A, B, Supplementary Data 1). The direct interaction between these proteins was further confirmed by a glutathione-S-transferase (GST)-binding assay using recombinant His-OTUD1 and GST-FGL1 in vitro (Fig. 4C). OTUD1 can regulate protein stability through deubiquitination[35]. We found that OTUD1 knockdown markedly increased the ubiquitination and degradation of FGL1 in MC38 and HT29 cells pretreated with Baf-A1 (Fig. 4D, E) and vice versa (Fig. 4F, G). Moreover, downregulation of OTUD1 abolished TAM-mediated deubiquitination of FGL1 and reduced TAM-induced FGL1 stabilization in MC38 and HT29 cells cultured with TAM-CM (Fig. 4H, I). Consistently, half-life analysis using cycloheximide pulse-chase analysis showed that downregulation of OTUD1 attenuated the protein half-life of FGL1 in tumor cells cultured with TAM-CM (Fig. 4J). Based on these observations, we further explored the molecular mechanisms underlying OTUD1-mediated deubiquitination of FGL1 by generating a catalytically inactive mutant OTUD1^C320S according to a previous report[35]. The results show that loss of OTUD1 deubiquitinase activity failed to decrease ubiquitination of FGL1 in 293T cells, indicating that the deubiquitination of FGL1 was dependent on OTUD1 deubiquitinase activity (Fig. 4K). In addition, we found that the ubiquitin-interacting motif (UIM) domain of OTUD1 was required for its binding to FGL1 (Fig. 4L, M). Additionally, incubation with TAM-CM decreased T-cell-mediated cell death of MC38-OVA cells, which could be rescued by knockdown of OTUD1 in these cells (Fig. 4N). Taken together, these data indicate that OTUD1 is involved in TAM-mediated FGL1 stabilization and immune evasion.

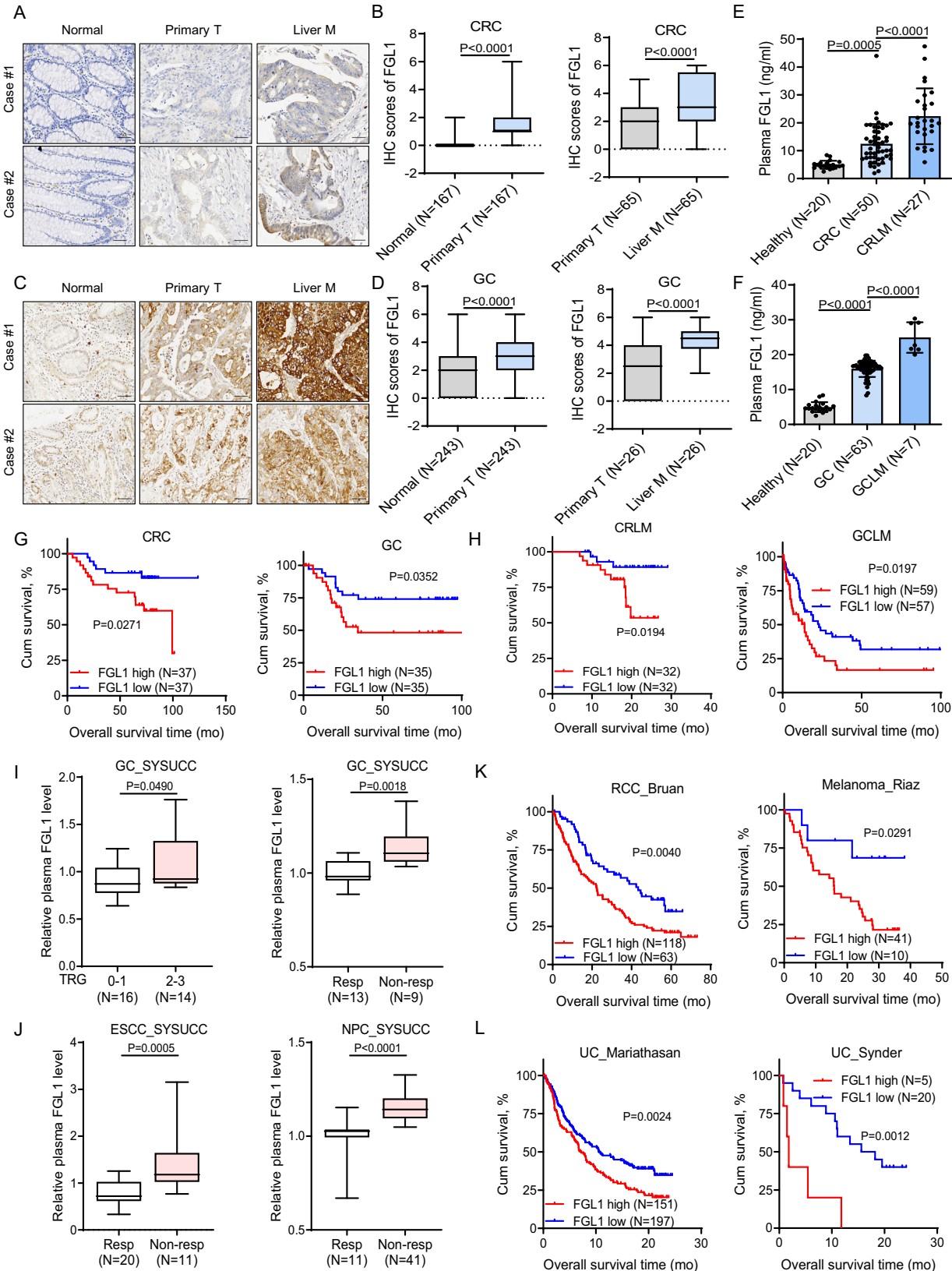

## OTUD1 is transcriptionally upregulated by the TAM/NF-κB signaling pathway

Overall, the results show that TAMs promoted FGL1 stability by activating the NF-κB/p65 signaling pathway; in addition, the deubiquitinase OTUD1 was involved in TAM-mediated stabilization of FGL1, but the intrinsic crosstalk between NF-κB/p65 and OTUD1 remains unclear.

NF-κB/p65 is a transcription factor that plays critical roles in inflammation, immunity, etc.[34] Therefore, we investigated whether OTUD1 expression is transcriptionally upregulated by TAM-mediated NF-κB/p65 activation. We first evaluated their expression correlation in TCGA gastrointestinal tumors and performed qPCR verification. The results reveal that *OTUD1* mRNA expression was highly correlated with *p65*

**Fig. 2 | High FGL1 levels predict poor outcomes and less benefit from PD-1/PD-L1 blockade therapy. A** Representative IHC staining of FGL1 in normal tissues, primary CRC tumor tissues and paired liver metastatic tissues. CRC, colorectal cancer. Scale bar, 50 μm. **B** The IHC staining scores of FGL1 in paired primary CRC tumors or liver metastatic tissues. **C** Representative IHC staining of FGL1 in normal tissues, primary GC tumor tissues and paired liver metastatic tissues. Scale bar, 50 μm. GC, gastric cancer. **D** The IHC staining scores of FGL1 in paired primary GC tumors or liver metastatic tissues. **E, F** ELISA detection of plasma FGL1 levels in healthy donor (N = 20), CRC (N = 50), CRLM (N = 27) specimens (SYSUCC cohort), GC (N = 63), and GCLM (N = 7) specimens (SYSUCC cohort). CRLM, colorectal cancer liver metastasis. GCLM, gastric cancer liver metastasis. **G** Kaplan–Meier analysis of the overall survival of CRC patients and GC patients with plasma FGL1 levels detected in (**E** and **F**). **H** Kaplan–Meier analysis of the overall survival of CRLM patients (N = 64, SYSUCC cohort) and GCLM patients (N = 116, SYSUCC cohort) with plasma FGL1 levels. **I, J** Plasma FGL1 level is inversely correlated with the anti-PD-1 blockade response as evaluated by the NCCN tumor regression grading system in neoadjuvant anti-PD-1 plus chemotherapy GC cohort (N = 30), anti-PD-1 plus chemotherapy GC cohort (N = 22) (**I**), anti-PD-1 plus chemotherapy advanced ESCC cohort (N = 31) and anti-PD-1 monotherapy advanced NPC cohort (N = 52) (**J**). ESCC, esophageal squamous cell carcinoma. NPC, nasopharyngeal carcinoma. **K** FGL1 expression is inversely correlated with overall survival in patients treated with anti-PD-1 (RCC_Bruan, N = 181, Melanoma_Riaz, N = 51). RCC, renal cell carcinoma. **L** FGL1 expression is inversely correlated with overall survival in patients treated with anti-PD-L1 (UC_Mariathasan, N = 348, UC_Synder, N = 25). UC, urothelial cancer. The data in (**B, D, I, J**) are presented as a box-and-whisker graph (min-max), and the horizontal line across the box indicates the median. The data in E and F are presented as the mean ± SD. The data in **G, H** and **K, L** were determined by Kaplan–Meier analysis with the log-rank test. P values were determined by two-tailed unpaired Student's t test (**B, D, I, J**) and one-way ANOVA (**E, F**).

expression in CRC and GC (Fig. 5A). This positive correlation was also validated in the Cancer Cell Line Encyclopedia (CCLE) database and CRC cell lines (Fig. 5B). Additionally, we analyzed chromatin immunoprecipitation sequencing (ChIP-seq) data in public datasets[36] and found that the OTUD1 promoter harbored p65 binding sites (Fig. 5C). OTUD1-promoter-luciferase (OTUD1-luc) activity was decreased by knockdown of p65 in MC38, HT29 and 293T cells analyzed by the luciferase reporter assay, and vice versa (Fig. 5D, E, S4A, B). We identified two binding elements of p65 in the 5′ flanking regions (from −2000 to +1) of both the human and mouse OTUD1 promoters (Fig. 5F), which was confirmed by ChIP-PCR assay (Fig. 5G). Furthermore, a luciferase reporter assay revealed that OTUD1 promoter activity was significantly increased in both MEF and 293T cells overexpressing phosphorylated (P)-p65 compared with the empty vector (Fig. 5H, I). The increased OTUD1 luciferase activity was diminished when two p65 binding sites were mutated concurrently, suggesting that two p65 binding sites function together to facilitate OTUD1 transcription (Fig. 5H, I). In addition, knockdown of p65 was sufficient to block TAM-mediated upregulation of OTUD1 transcriptional activity (Fig. 5J) and increase *OTUD1* mRNA and protein levels (Fig. 5K, L). Consistently, TAM-mediated upregulation of OTUD1 was also attenuated through blockade of the NF-κB signaling pathway by BAY 11−7082 (Fig. 5M). These data suggest that TAM-mediated NF-κB/p65 activation transcriptionally upregulates OTUD1.

## The TAM/TNFα-OTUD1-FGL1 axis promotes immune escape and progression of CRLM

We next examined the roles of the TAM/TNFα-OTUD1-FGL1 axis in liver metastasis in vivo. According to preliminary experiment, approximately one week after MC38 transplantation, there was a notable increase in plasma FGL1 levels, coinciding with the presence of established metastases and activated T cells (Fig. S5A−C). We found that injection with TNFα in late-stage mice with intraportal transplantation promoted metastatic tumor progression with increased liver weight and metastases (Fig. 6A, B), and such effects were accompanied by decreased IFN-γ⁺ CD8⁺/CD4⁺ and Ki67⁺ CD8⁺/CD4⁺ T-cell infiltration in the tumor microenvironment (Fig. 6C and S5D). Concurrently, the number of Treg cells in liver metastases was significantly increased, but no notable difference in Treg cell activity was observed (Fig. S5E, F). However, knockdown of either OTUD1 or FGL1 in MC38 cells reduced hepatic metastases, which could be rescued by overexpressing rFGL1, indicating the critical role of FGL1 in TNFα-mediated progression of metastatic tumors (Fig. 6A, B). Additionally, OTUD1 and FGL1 depletion significantly restored TNFα-reduced anticancer immunity and prolonged the OS of mice stimulated with TNFα (Fig. 6C, D, S5D), consistent with the above in vitro findings. In addition, we constructed intraportal transplantation mouse model using PBMC humanized mice and observed that co-injection of HT29 cells with TAMs significantly increased tumor burden in the liver

(Fig. S5G, H). Then we utilized an anti-CSF1R antibody to delete TAMs[37], and treated mice with anti-CSF1R antibody and/or anti-PD-1 antibody. Treatment with anti-CSF1R antibody alone markedly decreased metastatic tumor burden (Fig. 6E, F). However, the anti-PD-1 antibody had a limited effect on MC38 tumor growth in the liver microenvironment compared with the control IgG antibody (Fig. 6E, F), consistent with a previous study finding[38]. Interestingly, the combination of anti-CSF1R and anti-PD-1 treatment further significantly inhibited metastatic tumor progression in the liver and enhanced IFN-γ⁺ CD8⁺/CD4⁺ and Ki67⁺ CD8⁺/CD4⁺ T-cell infiltration in the tumor microenvironment (Fig. 6E, F, S5I). These results indicate that disruption of the TAM/TNFα-OTUD1-FGL1 axis in the liver microenvironment inhibits metastatic tumor progression by activating antitumor immunity, and may synergize with anti-PD-1 therapy.

We next investigated the clinical implications of plasma FGL1 levels in human CRLM tissues. Specifically, we tested the plasma FGL1 level of CRLM patients (from Fig. 2C, N = 27) and categorized samples into low-FGL1 and high-FGL1 groups according to the median level. We analyzed the correlation between the plasma FGL1 levels and CD68 (macrophage marker), P-p65, OTUD1, CD4, and CD8 by immunohistochemistry (IHC) in CRLM tissues (SYSUCC, N = 27) (Fig. 6G). Notably, the high-FGL1 group exhibited higher CD68, P-p65, and OTUD1 expression and lower CD4 and CD8 expression, whereas the low-FGL1 group showed the opposite pattern (Fig. 6H). In addition, we tested the association between the expression of CD68, TNFα, and OTUD1 and the efficacy of immunotherapy in published anti-PD-1/PD-L1 treatment cohorts[22,25]. We found that CD68, TNFα, and OTUD1 expression was inversely correlated with the overall survival of cancer patients (Fig. 6I, S5J). Our results indicate that the TAM/TNFα-OTUD1-FGL1 axis plays potential roles in cancer immunosuppression and the progression of metastatic tumors in the liver microenvironment.

## Benzethonium chloride inhibits tumor cell progression in the liver microenvironment by reducing the secretion of FGL1

To identify agents that can be used to inhibit secretion of FGL1, we exposed HT29 cells for 24 h to a compound library containing 1430 Food and Drug Administration (FDA)-approved drugs at 10 μM, followed by the detection of FGL1 secretion and normalization by cell viability. The pre- to posttreatment ratio of FGL1 for each compound was calculated, and hits were identified by z score analysis (Fig. 7A). We selected drugs with z scores greater than −3 (based on the 3-sigma rule) as candidates that decreased FGL1 levels[39,40] and identified that benzethonium chloride significantly inhibited the secretion of FGL1 in HT29, B16F10, MC38 cells and mouse hepatocytes (Fig. 7B, C). And we found that treated with benzethonium chloride had little effect on the viability of tumor cells (Fig. S6A). IB analysis showed that treatment with benzethonium chloride decreased the protein expression of FGL1 in cancer cells (Fig. 7D). Benzethonium chloride has surfactant, antiseptic and anti-infective properties[41]; however, its effects on metastatic

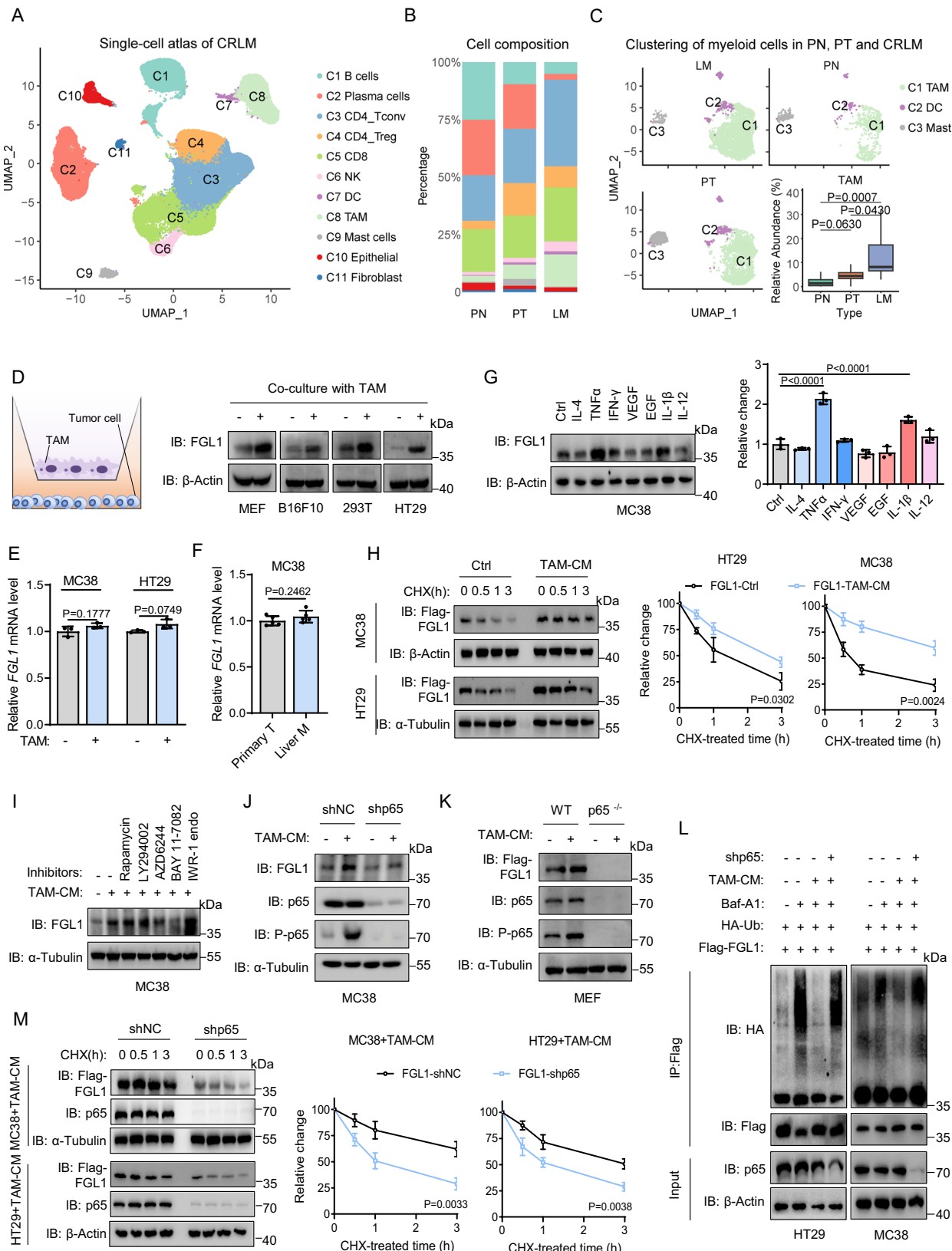

tumors have not yet been explored. We next tested its antitumor effects using an intraportal transplantation mouse model with MC38 or B16F10 cells and found that treatment with benzethonium chloride (3 mg/kg, 5 mg/kg) significantly decreased metastatic tumor burden without affecting host body weight and the number of TAMs in liver metastases (Fig. 7E, F and S6B–E). As expected, the decreased plasma

FGL1 levels in mice treated with benzethonium chloride were confirmed by ELISA (Fig. S6F). Since disruption of the TAM-OTUD1-FGL1 axis could be synergized with anti-PD-1 therapy, we also detected the synergistic antitumor effect of benzethonium chloride treatment and anti-PD-1 therapy (Fig. 7G, H, S6G, H). Interestingly, benzethonium chloride treatment synergized with anti-PD-1 therapy, leading to

**Fig. 3 | TAMs promote the stabilization of FGL1 by activating NF-κB in the liver microenvironment. A** UMAP embedding of transcriptional profiles from a previously published single-cell CRLM atlas (No. of cells = 98,072). **B** Stacked bar chart showing the proportion of cells in each cluster in primary normal tissue (PN), primary tumor tissue (PT), and liver metastatic tissue (LM). **C** UMAP embedding and relative abundance of myeloid cells in PNs ($n = 10$), PTs ($n = 10$), and LMs ($n = 10$). **D** IB detection of FGL1 expression in indicated cells cocultured with TAMs for 16 h. TAMs, tumor-associated macrophages. **E** qRT–PCR analysis of *FGL1* mRNA levels in indicated cells incubated with TAM, $n = 3$ biologically independent experiments. **F** qRT–PCR analysis of *FGL1* mRNA levels in MC38 cells sorted from primary and liver metastatic tumor tissues, $n = 5$ biologically independent experiments. **G** IB detection and quantification of FGL1 expression in MC38 cells stimulated with the indicated cytokines (10 ng/mL), $n = 3$ biologically independent experiments. **H** IB detection and quantification of FGL1 expression in indicated cells incubated with TAM conditioned medium (TAM-CM) following treatment with 40 μg/mL

cycloheximide (CHX) for the indicated times, $n = 3$ biologically independent experiments. **I** IB detection of FGL1 expression in MC38 cells incubated with TAM-CM and treated with the indicated inhibitors (10 μM) for 1 h. **J** IB detection of FGL1 expression in indicated cells incubated with TAM-CM. **K** IB detection of Flag-FGL1 expression in WT or *p65−/−* MEFs transiently overexpressing Flag-FGL1 following incubation with TAM-CM. **L** Coimmunoprecipitation analysis of the interaction between HA-Ub and Flag-FGL1 in indicated cells pretreated with Baf-A1 (12 h, 100 nM) and subjected to TAM-CM. **M** IB detection and quantification of Flag-FGL1 expression in indicated cells, $n = 3$ biologically independent experiments. The data in (**C**) are presented as a box-and-whisker graph (min-max), and the horizontal line across the box indicates the median. IB experiments in (**D**, **G**–**M**) were repeated three times, the data are representative of three biologically independent experiments. The data in (**E**–**G**, **H**, **M**) are presented as the mean ± SD. *P* values were determined by one-way ANOVA (**C**, **E**, **G**) and two-tailed unpaired Student's *t* test (**F**, **H**, **M**).

distinct inhibition of liver weight and a significant decrease in liver metastases (Fig. 7H and S6H). These effects were accompanied by significantly enhanced IFN-γ⁺ CD8⁺/CD4⁺ and Ki67⁺ CD8⁺/CD4⁺ T-cell infiltration in the tumor microenvironment compared with treatment alone (Fig. 7I–K, S6I, J). Consistently, the overall survival of mice receiving the combination therapy was significantly prolonged compared with that of mice in either monotherapy group (Fig. 7L and S6K). Collectively, these data suggest that the combination of benzethonium chloride and anti-PD-1 therapy has potential clinical implications in the treatment of liver metastases.

As indicated by the working model shown in Fig. 7M, our study revealed a model in which the liver microenvironment orchestrates FGL1-mediated immune escape and progression of liver metastatic CRC.

## Discussion

Emerging evidence indicates that the presence of liver metastasis is associated with poor response to immunotherapy, and the treatment of cancer patients with liver metastases needs to consider the distinct immunosuppressive phenotype in the liver microenvironment[3,6]. Mechanisms underlying the hepatic immunotolerant microenvironment include induction of regulatory T cells and dysfunction and exhaustion of effector T cells[42,43]. A previous study showed that inhibitory receptors, including TIM3, CTLA-4, PD-1 and LAG-3, were more highly expressed on tumor-infiltrating T cells in liver metastatic tumors[44]. LAG-3 is the most promising immune checkpoint next to PD-1 and CTLA-4 and is expressed on the surface of lymphocytes, such as CD4⁺ T cells and CD8⁺ T cells[45]. Blockade of LAG-3 enhanced tumor-infiltrating T-cell responses in mismatch repair-proficient liver metastasis of CRC[44]. Additionally, the FDA has approved the combination immunotherapy of anti-LAG-3 and anti-PD-1 as a first-line treatment for previously untreated metastatic melanoma[46]. In contrast, our present study focused on immunosuppressive molecules that are abnormally upregulated in liver metastatic tumor cells. Using different animal models and clinical tumor specimens, we demonstrated that FGL1 is highly expressed in liver metastases of gastrointestinal tumors and boosts tumor growth by inhibiting T-cell function. Although the LAG-3-FGL1 interaction mechanism remains unclear and controversial[47], our results are consistent with recent reports that FGL1, serving as one ligand of LAG-3[15], inhibits the tumor immune microenvironment by accelerating T-cell exhaustion and blocking T-cell proliferation. Thus, FGL1 has potential as another immune checkpoint target in clinical practice, especially in immunotherapy for liver metastatic tumor.

A complicated immune microenvironment within CRC liver metastases leads to an unsatisfactory prognosis[48]. Interactions between cancer cells and other cell populations contribute to the establishment of an immunosuppressive niche that promotes colonization and growth of CRC cells in the liver[26]. TAMs are a heterogeneous cell type and promote metastasis through the production of pro-

migratory factors and suppression of antitumor immunity[28,37]. A recent study showed that the liver microenvironment harbors reprogrammed suppressive macrophages, which were defined as MRC1⁺ CCL18⁺ macrophages, and these macrophages exhibit a sharp increase in metabolic activity in liver metastasis[5]. Another study revealed that liver metastases diminish immunotherapy efficacy by recruiting immunosuppressive CD11b⁺ F4/80⁺ macrophages that induce antigen-specific T-cell apoptosis within the liver microenvironment[7]. According to the literature, intercellular networking between macrophages and fibroblasts also supports CRC growth in the immunosuppressive metastatic niche in the liver[49]. In our data, accumulated TAMs in the liver microenvironment enhanced the stabilization of the immunosuppressive molecule FGL1 and promoted metastatic tumor progression. In line with previous studies[50,51], we also demonstrated that targeting TAMs in the metastatic tumor microenvironment using an anti-CSF1R antibody augmented the efficacy of anti-PD-1 therapy in liver metastasis. These data further imply that targeting TAMs and related signaling pathways is a potential strategy for liver metastasis treatment.

Posttranslational modifications (PTMs) exert subtle but profound influences on the activation, differentiation and functional fate of immune cells by regulating the stability and function of proteins[52,53]. Exploring PTMs in immunosuppressive molecules is a potential strategy to enhance antitumor immune responses[53,54]. For example, the expression of PD-L1 on tumor cells is regulated by various post-translational modifications, such as glycosylation, ubiquitination, palmitoylation, and phosphorylation[54]. Ubiquitination of endogenous proteins is a key regulatory step that guides protein degradation and crucial cell viability, differentiation, innate and adaptive immunity, etc.[55]. Ubiquitination and deubiquitination of PD-L1 regulate its proteasomal degradation and affect PD-1/PD-L1-mediated immunosuppression[34]. OTUD1 is a deubiquitinating enzyme involved in many cellular processes, including cancer and innate immune signaling pathways. Accumulating evidence suggests that OTUD1 is a critical regulator of gene transcription, cancer metastasis and chemoresistance[56–58]. A previous study showed that OTUD1 enhances iron transport and potentiates host antitumor immunity in colon cancer[35]. In this study, we uncovered that OTUD1 acts as a link between TAM-mediated NF-κB activation and FGL1-mediated immune escape by binding to FGL1 and deubiquitination events. Our findings add scientific content to understanding the regulation of the immune checkpoint molecule FGL1 and the functional roles of OTUD1-mediated deubiquitination in cancer immunology.

Liver metastasis has a distinct immune environment and requires specific diagnostic markers and immunotherapeutic strategies. The FGL1 expression level can be a marker of poor prognosis in various tumors[11]. For instance, high plasma FGL1 levels are associated with worse OS in NSCLC and metastatic melanoma patients treated with anti-PD-1 mAbs[15]. Consistently, our results show that high FGL1 levels predict poor outcomes and less benefit from PD-1/PD-L1 blockade

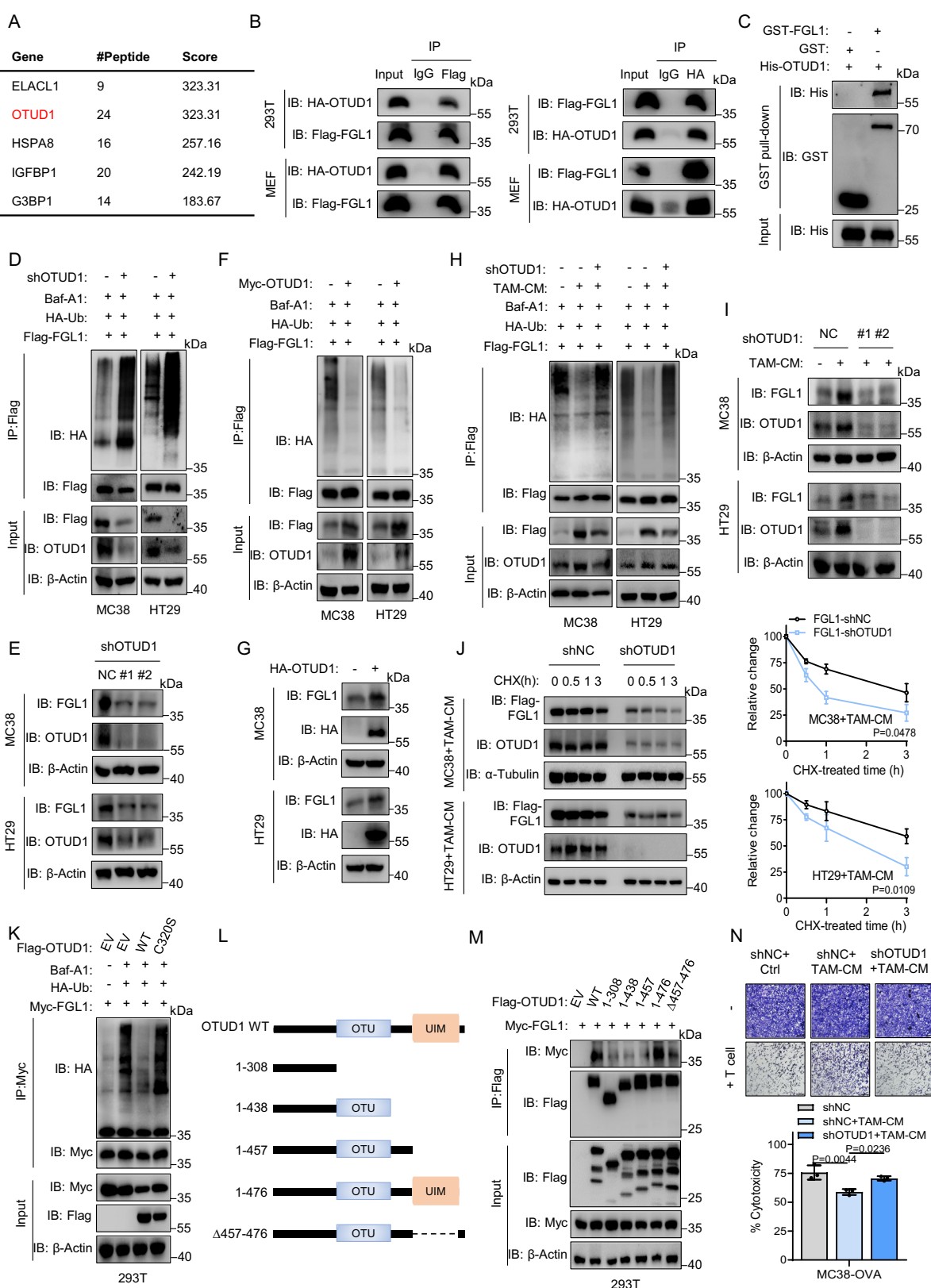

therapy in patients with gastroenterological cancer. In addition, an increased understanding of the liver microenvironment will facilitate the development of immunotherapies for liver metastasis[4,5]. A previous study showed that liver-directed radiotherapy eliminates immunosuppressive hepatic macrophages and that combination with immunotherapy could promote systemic antitumor immunity[7].

Another study revealed that T cells in liver metastasis produce CCL5 to stimulate protumoral effects, and blockade of CCR5 shows anti-tumoral effects in CRC patients with liver metastases[59]. Although there are currently no mAbs targeting FGL1, many mAbs targeting LAG-3 are undergoing clinical trials, including relatlimab (first LAG-3 mAb), and show good potential in tumor immunotherapy[60]. Notably, recent

**Fig. 4 | The deubiquitinase OTUD1 is involved in TAM-mediated stabilization of FGL1. A** Mass spectrometry (MS) analysis to explore different FGL1-binding proteins in 293T cells cocultured with TAMs. The representative candidates are listed. OTU deubiquitinase 1 (OTUD1) was identified. **B** Co-IP analysis of the interaction between exogenously overexpressed HA-OTUD1 and Flag-FGL1 in 293T and MEF cells. **C** GST pull-down assay analysis of the interaction between His-OTUD1 and GST-FGL1 proteins in vitro. **D** Co-IP analysis of the interaction between HA-Ub and Flag-FGL1 in OTUD1 knockdown or control MC38 and HT29 cells pretreated with Baf-A1 (100 nM, 24 h). **E** IB detection of FGL1 and OTUD1 expression in indicated cells. **F** Co-IP analysis of the interaction between HA-Ub and Flag-FGL1 in Myc-OTUD1-overexpressing or control MC38 and HT29 cells pretreated with Baf-A1 (100 nM, 24 h). **G** IB detection of FGL1 and HA-OTUD1 expression in indicated cells. **H** Co-IP analysis of the interaction between HA-Ub and Flag-FGL1 in indicated cells pretreated with Baf-A1 (100 nM, 12 h), and then incubated with TAM-CM for 16 h. **I** IB detection of FGL1 and OTUD1 expression in indicated cells incubated with TAM-

CM. **J** IB detection and quantification of FGL1 expression in the indicated cells incubated with TAM-CM for 16 h following treatment with CHX for different times, $n = 3$ biologically independent experiments. **K** Co-IP analysis of the interaction between HA-Ub and Myc-FGL1 in 293T cells overexpressing Flag-OTUD1, Flag-OTUD1 (C320S) or control (EV). **L** Schematic diagrams of OTUD1 displaying the positions of different domains. **M** Co-IP analysis of the interaction of different truncations of Flag-OTUD1 with Myc-FGL1 in 293T cells. **N** Representative images and quantification of the indicated cells cocultured with activated OT-1 T cells for 8 h and then subjected to crystal violet staining, $n = 3$ biologically independent experiments. The ratio of MC38-OVA cells to OT-1 T cells was 5:1. IB experiments in (**B**–**K**, **M**) were repeated three times, the data are representative of three biologically independent experiments. The data in **J**, **N** are presented as the mean ± SD. The $P$ value was determined by one-way ANOVA (**N**) and two-tailed unpaired Student's $t$ test (**J**).

---

clinical trial results showed that combining relatlimab with nivolumab (anti-PD-1 mAb) provides a satisfying benefit for patients with unresectable melanoma or patients with resectable melanoma as neoadjuvant therapy[46,61]. Importantly, the combined therapy was also far less toxic to patients. In addition, oxysophocarpine has been reported to enhance the therapeutic effect of anti-LAG-3 mAb in HCC by inhibiting the expression of FGL1[62]. In this study, we uncovered that benzethonium chloride also synergizes with anti-PD-1 therapy in liver metastatic tumor models by reducing the secretion of FGL1, providing a preclinical basis for the clinical application of benzethonium chloride.

Taken together, our study enriches the understanding of the mechanistic connections among immune cells (i.e., TAMs and T cells) and tumor cells in the liver microenvironment that facilitate immune escape and CRC progression. In addition, this study elucidates the critical roles and regulatory mechanisms of FGL1 in promoting metastatic tumor cell progression, emphasizing its prognostic value and the clinical significance of immunotherapy. Given that inhibition of FGL1 synergizes with PD-1 blockade in mouse models, further investigation of the combination of anti-FGL1 mAbs and ICB therapy may shed light on strategies for liver metastatic cancer immunotherapy.

## Methods

### Ethical statement

All patient samples were obtained from Sun Yat-sen University Cancer Center (SYSUCC, Guangzhou, China). All samples were collected after obtaining written informed consent and in accordance with the guidelines of the Medical Ethics Committee of SYSUCC and the Declaration of Helsinki. This study was approved by the Ethical Committee of Sun Yat-sen University Cancer center (2022112917400000300289). All animal studies were performed in accordance with a protocol approved by the Institutional Ethics Committee for Clinical Research and Animal Trials of the SYSUCC (L025501202112009).

### Cell lines

B16F10, CT26, 293T, and HT29 cells were purchased from American Type Culture Collection (ATCC, Manassas, USA). The MC38 (mouse colon adenocarcinoma) cell line was kindly gifted by Prof. Wei Yang (Guangdong Provincial People's Hospital). MEF WT, MEF $p65^{-/-}$, and MEF $IKK^{-/-}$ cells were kindly provided by Dr. Paul J. Chiao's laboratory (MD Anderson Cancer Center, USA)[63]. The cells were cultured in RPMI-1640 or DMEM (Gibco, Grand Island, USA) supplemented with 10% FBS (Wisent Inc, Quebec, Canada) and 1% penicillin/streptomycin (Gibco) at 37 °C with 5% $CO_2$. All cells were authenticated via short tandem repeat (STR)-PCR DNA profiling, and were determined to be free of mycoplasma contamination.

### RNA isolation and qPCR analysis

RNA from cells and tissues was extracted using TRIzol Reagent (Thermo Scientific, Carlsbad, USA) according to the manufacturer's

instructions. Next, complementary DNA was synthesized using the Prime Script RT Master Mix Kit (Takara, Tokyo, Japan) and served as a template for real-time PCR using the GoTaq qPCR Master Mix Kit (Promega, Madison, USA) according to the manufacturer's instructions. β-Actin was used as the internal control gene, and relative gene expression was analyzed using the $2^{-\Delta Ct}$ or $2^{-\Delta\Delta Ct}$ method. The sequences of the primers are listed in Supplementary Data 2.

### FGL1 detection by ELISA

FGL1 concentrations in human/mouse plasma and supernatant were determined using the Human FGL1 ELISA kit (Elabscience Biotechnology, Wuhan, China) and Mouse FGL1 ELISA kit (Fine Test, Wuhan China). Standard curve calibrators, plasma samples, and supernatant samples were assayed according to the manufacturer's instructions.

### Ligand binding assay

Recombinant hLAG-3 (1 μg/mL) (ABclonal Technology, Wuhan, China) was coated on ELISA microwells, and then different amounts of recombinant hFGL1 (ABclonal Technology, Wuhan, China) were incubated for 1 h. The hFGL1 binding was finally detected using an HRP-conjugated anti-hFGL1 antibody. OD, optical density (450 nm).

### CD8 + T-cell cytotoxicity

MC38-OVA cells were seeded into 6-well plates overnight. OT-1 CD8+ T cells were isolated from the spleens of OT-1 mice and treated with 10 ng/mL IL-2 (PeproTech, Rocky Hill, USA), 10 nM T-Select H-2Kb OVA Tetramer (OVA 257–264) (Medical & Biological Laboratories, Nagoya, Japan) and 100 mM β-mercaptoethanol (Sigma Aldrich, Steinheim, Germany) for 3–5 days. For experiments involving TAM coculture, MC38-OVA cells were precultured with TAM-CM for 16 h. Purified OT-1 CD8+ T cells were cocultured with shRNA-modified MC38-OVA cells or pretreated MC38-OVA cells at a ratio of 1:5. After 8 hr, the tumor cells were gently washed and subjected to crystal violet staining (Beyotime, Shanghai, China).

### T-cell function assays

A T-cell costimulation assay was conducted as previously described[15]. Briefly, splenocytes isolated from C57BL/6 J mice were treated with 2.5 μg/mL anti-CD3ε mAb (Biolegend, San Diego, USA), 1.25 μg/mL anti-CD28 mAb (Biolegend) and 5 ng/mL recombinant mouse IL-2 (PeproTech) for 12 h. Thereafter, the pretreated splenocytes were seeded into 96-well plates (NEST Biotechnology, Wuxi, China) at $5 \times 10^4$/well and incubated with recombinant murine FGL1 (10 μg/mL), anti-LAG-3 antibody (BioXcell, Lebanon, USA) or control for 72 h. The supernatant was collected, and the IFN-γ levels were determined using a Mouse IFN-γ Valukine ELISA Kit (R&D, Minneapolis, USA) according to the manufacturer's instructions.

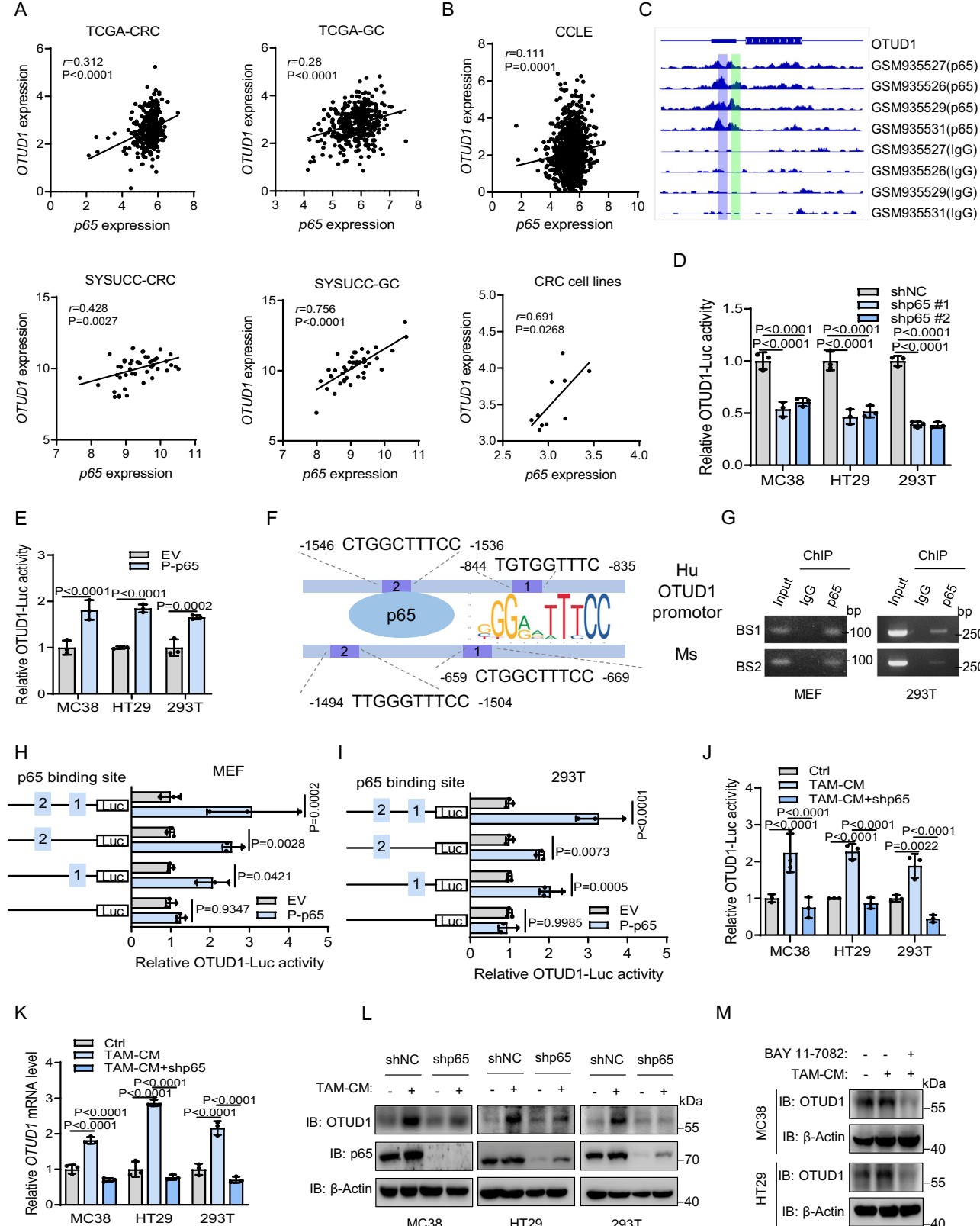

## Cell transfection and lentiviral-based gene transduction

Human or mouse Flag-OTUD1, HA-OTUD1, Flag-FGL1, Myc-FGL1, human Flag-OTUD1$^{C320S}$, Flag-OTUD1$^{1-308}$, Flag-OTUD1$^{1-438}$, Flag-OTUD1$^{1-476}$, Flag-OTUD1$^{\Delta457-476}$, and Flag-P-p65 were cloned into the pCDNA3.1 vector (GeneCopoeia, Rockville, USA). The plasmids were transiently transfected into cells using Lipofectamine 3000 Reagent (Thermo Scientific)

according to the manufacturer's instructions. The lentiviruses for mouse FGL1 overexpression, mouse shFGL1 expression, human shFGL1 expression, human FGL1 overexpression, human shOTUD1 expression and recombinant adeno-associated virus for mouse shFGL1 expression were synthesized by OBiO Technology (Shanghai, China). The shRNA of FGL1 was constructed into pAAV-TBG-MCS by OBiO Technology

**Fig. 5 | OTUD1 is transcriptionally upregulated by the TAM/NF-κB signaling pathway. A** Correlation between *p65* and *OTUD1* expression in the TCGA-CRC (*N* = 376) and TCGA-GC (*N* = 370) datasets and the cohort of CRC (*N* = 47) and GC (*N* = 44) patients in SYSUCC data. The Pearson correlation coefficient (r) and *P* value are displayed. **B** Correlation between *p65* and *OTUD1* expression in the Cancer Cell Line Encyclopedia (CCLE) dataset (*N* = 1040) and CRC cell lines (*N* = 10). **C** The p65 ChIP-seq signals surrounding the OTUD1 gene locus in the indicated datasets. **D** Luciferase activity of the OTUD1 promoter reporter in p65 knockdown or control (shNC) MC38, HT29 and 293T cells, *n* = 3 biologically independent experiments. **E** Luciferase activity of the OTUD1 promoter reporter in P-p65-overexpressing or control (EV) MC38, HT29 and 293T cells, *n* = 3 biologically independent experiments. **F** The OTUD1 promoter of human and mice contains consensus p65-binding regions. **G** Chromatin immunoprecipitation (ChIP)-PCR analysis of the interaction between p65 and the OTUD1 promoter in MEFs and 293T cells. **H**, **I** Luciferase activity of the OTUD1 promoter reporter with the indicated p65 binding site-mutated OTUD1 in P-p65-overexpressing or control (EV) MEFs (**H**) and 293T cells (**I**), *n* = 3 biologically independent experiments. **J** Luciferase activity of the OTUD1 promoter reporter in p65 knockdown or control (shNC) MC38, HT29 and 293T cells incubated with TAM-CM, *n* = 3 biologically independent experiments. **K** qRT–PCR analysis of *OTUD1* mRNA levels in p65 knockdown or control (shNC) MC38, HT29 and 293T cells incubated with TAM-CM, *n* = 3 biologically independent experiments. **L** IB detection of OTUD1 and p65 in the indicated cells incubated with TAM-CM. **M** IB detection of OTUD1 in MC38 and HT29 cells pretreated with BAY 11–7082 (10 μM) for 1 h, followed by incubation with TAM-CM for 16 h. ChIP-PCR experiments in (**G**), IB experiments in (**L**, **M**) were repeated three times, the data are representative of three biologically independent experiments. The data in (**D**, **E**, **H**–**K**) are presented as the mean ± SD. *P* values were determined by two-tailed Pearson's chi-squared test (**A**, **B**) and one-way ANOVA (**D**, **E**, **H**–**K**).

(Shanghai, China). The plasmids for human/mouse shp65 expression and mouse shOTUD1 expression were purchased from GeneCopoeia (Rockville, USA). To produce lentivirus, 2.8 μg of the packaging plasmid psPAX2, 1.56 μg of the envelope vector pMD2. G and 2 μg of our above target plasmids were cotransfected into 293T cells cultured in T25 flasks using Lipofectamine 3000 Reagent (Thermo Scientific) according to the manufacturer's instructions. After 72 h, the medium containing virus was collected and passed through a 0.45 μm filter. Cells at a density of 70% in a 6-well plate were transduced with virus and 8 μg/mL polybrene (MedChemExpress, Shanghai, China)-containing medium. Viruses were removed 24 h after transduction, and cells were selected using puromycin (Solarbio, Beijing, China) for one week.

## IB and IHC assays

IB and IHC were performed according to the standard procedure as previously reported[29]. Briefly, cells or tissues were lysed in cell lysis buffer (Beyotime). The protein concentrations of cleared lysates were determined with a BCA assay kit (Thermo Scientific). Blotting membranes were stripped and reprobed with anti-β-Actin antibody as a loading control. Quantification of IB was analyzed by ImageJ software (National Institutes of Health, Bethesda, USA). For the IHC assays, stained IHC sections were incubated for 2 h at 37 °C with the primary antibody and then reviewed and scored independently by two expert pathologists. The final scores were determined by the intensity (0, 1, 2, 3) and proportion of positive cells (0–100%). Detailed information on the antibodies used is listed in Supplementary Data 3.

## Mass spectrometry analysis of FGL1-interactome

Immunoprecipitated Myc-FGL1 using anti-Myc antibody (DIA-AN, Wuhan, China) from 293T cells cocultured with tumor-associated macrophages for 16 h. The bound proteins were extracted from IP beads using SDT lysis buffer (4% SDS, 100 mM DTT, 100 mM Tris-HCl pH8.0) and prepared with in-solution trypsin digestion. The peptide was desalted with C18 StageTip for further LC-MS analysis.

LC-MS/MS experiments were performed on a Q Exactive HF-X mass spectrometer that was coupled to Easy nLC1200 (Thermo Scientific) by Shanghai Bioprofile Technology Company Ltd. (Shanghai, China). The mass spectrometer was operated in data-dependent mode with the following acquisition cycle: an MS scan (m/z 350–1800) recorded at resolution R = 60,000 and MS/MS scans recorded at resolution R = 15,000, which were acquired by HCD (higher-energy C-trap dissociation fragmentation) with collision energy of 28. MS data were searched with MaxQuant (v1.6.1.0, https://www.maxquant.org/). Top 30 candidate proteins are listed in Supplementary Data 1. All candidate proteins are deposited in https://doi.org/10.6084/m9.figshare.24104007, and the raw mass spectrometry proteomics data have been deposited to the ProteomeXchange Consortium (https://proteomecentral.proteomexchange.org/cgi/GetDataset?ID = PXD045948) via the iProX partner repository with the dataset identifier PXD045948.

## GST pull-down assay

GST pull-down assays were performed using Glutathione Sepharose 4B beads (GE Healthcare, Little Chalfont, UK). The 100 μL GST beads were washed with PBST three times, and 100 μg GST control or purified GST-FGL1 protein (Genecreate, Wuhan, China) was added and incubated at 4 °C for 1 h. Purified His-OTUD1 protein (100 μg, Genecreate) was incubated with the GST-FGL1-bead complex at 4 °C for 2 h. The incubated proteins were then washed using PBST five times, eluted by boiling in 1 × loading buffer and then analyzed by IB assay.

## Chromatin immunoprecipitation (ChIP) assay

The ChIP assay was performed using the MAGnify Chromatin Immunoprecipitation System (Thermo Scientific) according to the manufacturer's instructions. Briefly, $1 \times 10^7$ 293T or MEF cells were washed and resuspended in 1 mL PBS. Then, 27 μL of 37% formaldehyde was added and incubated with rotation for 10 min at room temperature. Thereafter, 114 μL of room-temperature 1.25 M glycine were added to stop the reaction. The cross-linked cells were washed twice with ice-cold PBS and lysed using 200 μL lysis buffer supplemented with 1 μL protease inhibitors on ice for 30 min. Next, the cell lysate was diluted using 800 μL dilution buffer supplemented with 4 μL protease inhibitors and sonicated on ice. The chromatin was sheared into 200–500 bp fragments and tested using a 1.5% agarose gel. Then, the sheared DNA mixture was subjected to IP with 1 μg p65 antibody or normal rabbit IgG overnight at 4 °C. The protein–DNA complexes were washed twice with IP buffer and treated using 53 μL reverse cross-linking buffer supplemented with 1 μL proteinase K at 55 °C for 45 min, then incubated at 65 °C for 45 min followed by incubation on ice for 5 min. Thereafter, the uncrosslinked DNA was purified, washed and eluted for PCR analysis.

## Luciferase assay

The wild-type, p65 binding site mutant pGL4.10-OTUD1-Luc plasmids of mouse/human and pGL4.10 empty plasmid were purchased from GeneCopoeia (Rockville, USA). Cells were transfected using Lipofectamine 3000 Reagent (Thermo Scientific) according to the manufacturer's instructions. pGL4.10-TK (GeneCopoeia) was cotransfected as an internal control to normalize the transfection efficacy. After transfection and experimental treatments, the cells were lysed, and the luciferase activity was measured using the Dual-Luciferase Reporter Assay System (Promega) according to the manufacturer's instructions. Protein expression from the luciferase assay was determined in the remaining cell lysate by IB assay.

## Animals

All animal studies were approved by the Institutional Ethics Committee for Clinical Research and Animal Trials of the SYSUCC. Mice were housed in temperature-controlled pathogen-free conditions (around 20 °C, 40% humidity) under a 12-h light/dark cycle. The randomization of animal allocation was performed by random number generation by

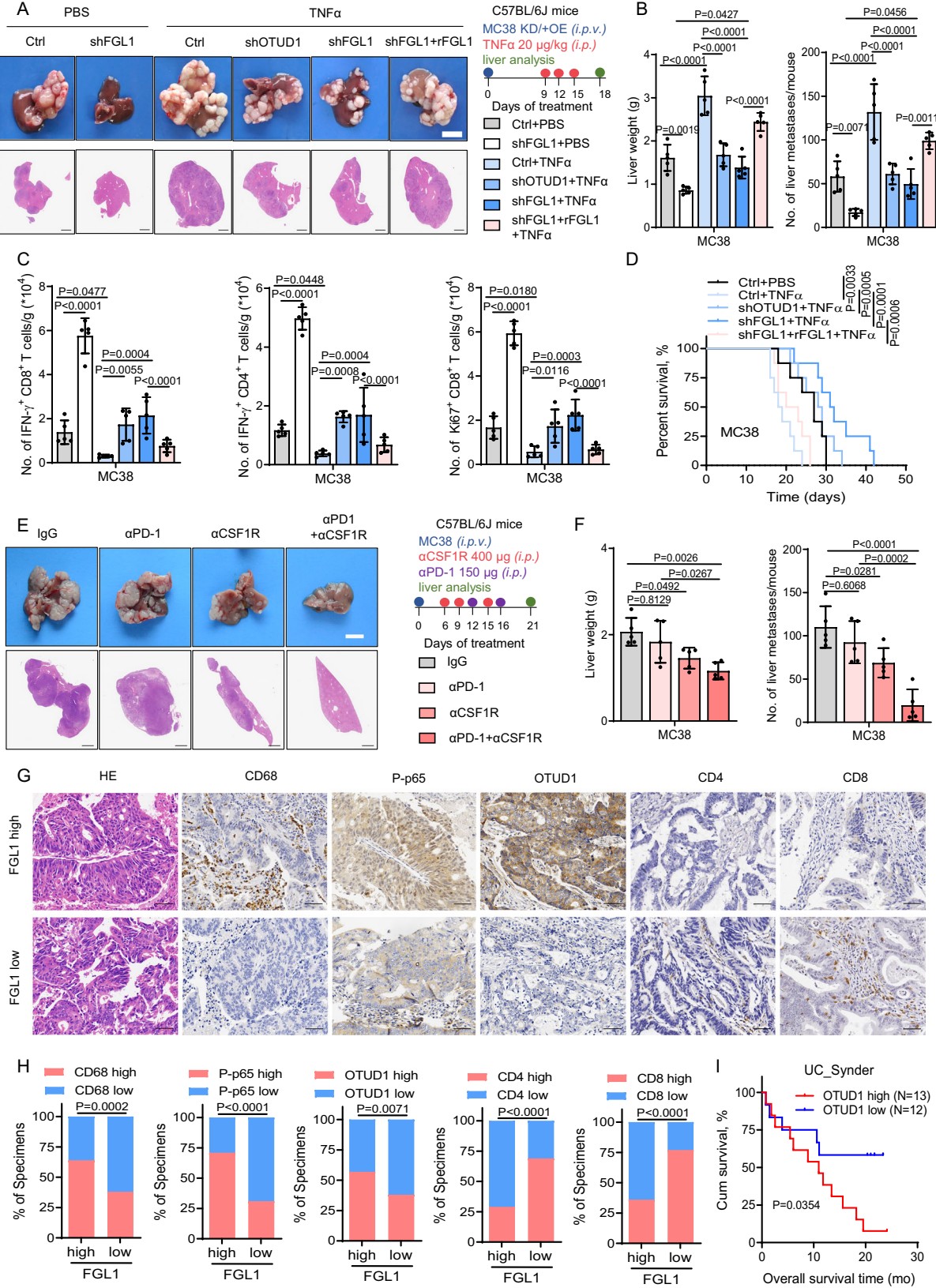

computer. For metastasis experiments relying on the tumor burden or liver surface metastasis number, the animal numbers were estimated based on previous experience with those models[23]. Pain and distress were monitored by observing the presence of rapid weight loss, weight loss exceeding 20% of body weight, hunched posture, lethargy, lack of movement, and rapid growth of tumor masses. Mice exhibiting any of these signs were euthanized by cervical dislocation. Transplanted tumors were not to exceed a diameter of 2.0 cm or 10% of body weight as permitted by the Institutional Ethics Committee for Clinical Research and Animal Trials of the SYSUCC. Since our experimental design did not involve gender-related factors, and our preliminary experiments showed that FGL1 knockdown phenotypes were identical

**Fig. 6 | The TAM/TNFα-OTUD1-FGL1 axis promotes immune escape and progression of CRLM. A** Representative H&E staining of metastatic livers from C57BL/6 J mice intraportally implanted with the control (Ctrl), OTUD1 knockdown, FGL1 knockdown or FGL1 knockdown with overexpression of short hairpin RNA (RNA)-resistant FGL1 (rFGL1) MC38 cells followed by TNFα treatment. Scale bar for bright-field images, 1 cm; scale bar for H&E images, 2.5 mm. Right panel, experimental strategies. **B** Quantification of liver weight and the number of liver metastases of the indicated groups of mice ($n = 5$ mice per group). **C** Flow cytometric analysis of the number of IFN-γ$^+$ CD8$^+$, IFN-γ$^+$ CD4$^+$ and Ki67$^+$ CD8$^+$ T cells in liver metastases of indicated groups of mice ($n = 5$ mice per group). **D** Survival curve analysis of mice in indicated groups (Mean survival times of Ctrl+PBS, Ctrl+TNFα, shOTUD1+TNFα, shFGL1+TNFα and shFGL1 + rFGL1+TNFα mice were 25.8, 19.1, 27.6, 32.5 and 21.5 days, respectively) ($n = 8$ mice per group). **E** Representative H&E staining of metastatic livers from C57BL/6 J mice intraportally injected with MC38 cells followed by treatment with the control (IgG), anti-CSF1R and/or anti-PD-1 monoclonal antibodies. Scale bar for bright-field images, 1 cm; scale bar for H&E images, 2.5 mm. Right panel, experimental strategies. **F** Quantification of liver weight and the number of liver metastases of indicated groups of mice ($n = 5$ mice per group). **G** Representative H&E and IHC staining of the indicated markers in CRC metastatic liver tissues from patients with high ($N = 14$) or low ($N = 13$) plasma FGL1 levels detected in Fig. 2 (**C**) ($N = 27$). Scale bar, 50 μm. **H** Percentage of specimens with high or low CD68, P-p65, OTUD1, CD4, and CD8 expression in the low or high plasma FGL1 groups. **I** OTUD1 expression is inversely correlated with overall survival in patients treated with anti-PD-L1 in the study by Snyder and colleagues on urothelial cancer (UC_Snyder, $N = 25$). The data in (**B**, **C**, **F**) are presented as the mean ± SD. The data in (**D**, **I**) were determined by Kaplan–Meier analysis with the log-rank test. The data in (**H**) were determined by the two-sided chi-square test. $P$ values were determined by one-way ANOVA (**B**, **C**, **F**).

in male and female mice, only female mice were used for the following experiments to control variables. All mice used in this study were C57BL/6 J female mice (6 weeks), which were purchased from Beijing Vital River Laboratories (Beijing, China), $Rag1^{-/-}$ (Strain NO. T004753) female mice (5–6 weeks) and NOD/ShiLtJGpt-Prkdc$^{em26Cd52}$Il2rg$^{em26Cd22}$/Gpt (NCG, Strain NO. T001475) female mice (5–6 weeks) that were purchased from Gempharmatech (Jiangsu, China). For the experimental liver metastasis mouse models, $3.5 \times 10^5$ shRNA-modified control or target–gene-knockdown tumor cells were injected into the portal vein of C57BL/6 J mice and $Rag1^{-/-}$ mice as reported previously[64]. Briefly, a 30 G syringe was used to inject 100 μL of cell suspension directly into the portal vein. For the AAV8-mediated liver transduction group, C57BL/6 J mice were used, after hepatic portal vein injection of MC38 cells ($3.5 \times 10^5$), AAV-shFGL1 or AAV-shNC (OBiO Technology, Shanghai, China) was injected intravenously on Day 2 because the preliminary test and previous studies show that the knockdown effect takes approximately two weeks to achieve[20]. For the TNFα treatment group, C57BL/6 J mice were used, 20 μg/kg TNFα (PeproTech) was injected intraperitoneally on Days 9, 12, and 15. For the PBMC humanized mice, NOD/ShiLtJGpt-Prkdc$^{em26Cd52}$Il2rg$^{em26Cd22}$/Gpt (NCG) mice were used, as we previously described[21], $1 \times 10^7$ hPBMCs (Xinjin Biotechnology Co., Ltd., Guangzhou, China) were injected intravenously, then HT29 cells ($1 \times 10^6$) or HT29 cells ($1 \times 10^6$) and CD14$^+$ TAMs ($1 \times 10^6$) were injected into the portal vein of PBMC humanized mice. In experiments involving anti-CSF1R antibody and anti-PD-1 antibody, 400 μg anti-CSF1R antibody (BioXcell, Lebanon, USA) or isotype control (BioXcell) was given on Days 6, 9, and 15. In addition, 150 μg of anti-PD-1 (Junshi Biosciences, Shanghai, China) or isotype control (Junshi Biosciences) was injected intraperitoneally on Days 12 and 16 in combination with anti-CSF1R. In experiments involving benzethonium chloride treatment, MC38 ($3.5 \times 10^5$) or B16F10 cells ($3.5 \times 10^5$) were injected into the portal vein of C57BL/6 J mice. 3 mg/kg or 5 mg/kg benzethonium chloride (Selleck Chemicals, Houston, USA) or PBS were injected intraperitoneally on Days 4, 6, 8, 10, 12, and 14. In experiments involving benzethonium chloride and anti-PD-1 antibody, MC38 ($3.5 \times 10^5$) or B16F10 cells ($3.5 \times 10^5$) were injected into the hepatic portal vein of C57BL/6 J mice. Benzethonium chloride (3 mg/kg) or PBS was injected intraperitoneally on Days 6, 10, 12, and 16. In addition, 150 μg of anti-PD-1 or isotype control was injected on Days 8 and 14 in combination with benzethonium chloride.

## Tissue Dissociation and flow cytometric analysis
Fresh isolated tumor samples from liver metastases were dissociated as previously described[21]. Briefly, the samples were minced into small pieces using a scalpel, digested with a Tumor Dissociation Kit (Miltenyi Biotec, Bergisch Gladbach, Germany) and incubated at 37 °C with rotation for 30 min. The samples were passed through 70 μm and 40 μm nylon cell strainers and washed with FACS buffer (PBS supplemented with 2% FBS). After removal of the supernatant, the remaining cells were gently suspended in 40% Percoll (Cytiva, Marlborough USA) and purified by gradient centrifugation for 20 min. The remaining cells were resuspended in FACS buffer, stained with Zombie for 15 min at room temperature, washed and stained with anti-mouse CD45, CD3, CD4, CD8 antibodies. The cells were then washed and fixed and permeabilized with Foxp3/Transcription Factor Staining Buffer (Thermo Scientific) overnight at 4 °C. Next, the fixed cells were washed with 1x intracellular staining perm and wash buffer and incubated with anti-mouse IFN-γ and/or Ki67 antibodies for 30 min at room temperature. Then washed cells were evaluated by flow cytometry and the data were analyzed using CytExpert software (Beckman Coulter, Brea, USA). To detect regulatory T cells, the remaining cells were resuspended in FACS buffer, stained with anti-mouse CD45, CD3, CD4, CD25, CTLA-4 and Foxp3 antibodies and evaluated by flow cytometry as described above. Detailed information of the antibodies used is listed in Supplementary Data 3.

## Isolation of primary mouse hepatocytes
Primary mouse hepatocytes were isolated using a two-step collagenase perfusion technique as previously described[16]. Isolated primary mouse hepatocytes were used for immunoblotting or seeded into 6-well plates at $1 \times 10^6$/well and cultured in William's E Medium (Gibco) supplemented with 10% FBS (Wisent Inc), 1% penicillin/streptomycin (Gibco), 0.25 μg/mL amphotericin B (Gibco), 1% insulin-transferrin-selenium-ethanolamine (Gibco) and 5 nM dexamethasone (Sigma Aldrich) at 37 °C with 5% CO2.

## Isolation of TAMs from liver metastasis
For human liver metastasis, TAMs were isolated as previously described[29]. Briefly, fresh tumor samples were sliced into 1–2 mm$^3$ pieces and digested with DMEM containing 1 mg/mL collagenase IV (Sigma Aldrich) and 5 U/mL DNase I (Thermo Scientific) at 37 °C with rotation for 30 min. Dissociated cells were filtered through 40 μm nylon cell strainers and washed with FACS buffer (PBS supplemented with 2% FBS). After filtration, the red blood cells were lysed using red cell lysis solution (Biosharp, Hefei, China). The remaining cells were washed and resuspended in FACS buffer. Thereafter, macrophages were isolated using the EasySep Human CD14 Positive Selection Kit II (StemCell Technologies, Vancouver, Canada) according to the manufacturer's instructions.

For mouse liver metastasis, freshly isolated tumor samples were dissociated into single cells as described above. From the resuspended single cells, F4/80+ cells were isolated using Anti-F4/80 MicroBeads UltraPure (Miltenyi Biotec) according to the manufacturer's instructions. Then, the macrophages were seeded ($5 \times 10^5$ cells/well) on 0.4 μm transwell inserts in 6-well plates. Next, the insert was transferred to a 6-well plate seeded with tumor cells. To prepare conditional media (CM) from tumor macrophages, $1 \times 10^6$ macrophages were resuspended in 2 mL of DMEM supplemented with 10% FBS and 1% penicillin/streptomycin and cultured. After 24 h, the supernatants were harvested repeatedly, centrifuged, and stored at −80 °C.

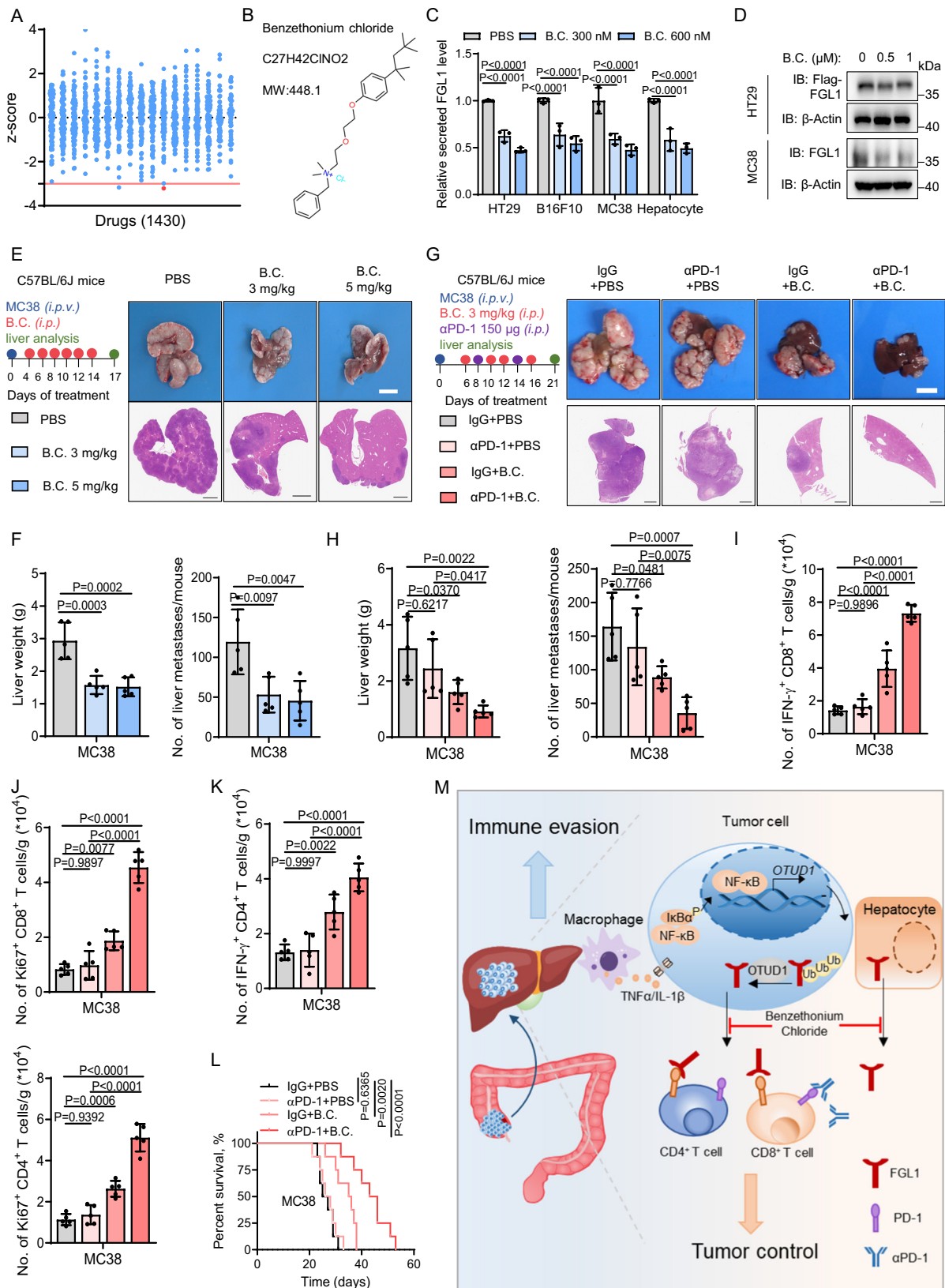

HT29 cells were seeded in 96-well plates and allowed to adapt for 24 h. Thereafter, 1430 drugs from the FDA-approved drug screening library (Selleck Chemicals) were added at a final concentration of 10 μM. Each plate included 3 sulfoxide (DMSO) controls. Cells were then incubated for 24 h, and the supernatants were collected to determine the FGL1 concentration using ELISA. Next, MTS assays were performed to test cell viability. The postdrug FGL1 concentrations were normalized to the cell viability, and the delta-FGL1 ratio (postdrug [FGL1]: postDMSO [FGL1]) was calculated. Then, a $z$ score was generated for each

**Fig. 7 | Benzethonium chloride inhibits tumor cell progression in the liver microenvironment by reducing the secretion of FGL1. A** Graph of the *z* scores (red line = −3) showing the effect of 1430 FDA-approved drugs (blue spheres) on the secretion of FGL1 in HT29 cells. **B** Structure of benzethonium chloride. **C** ELISA detection of FGL1 secreted by indicated cells treated with benzethonium chloride, *n* = 3 biologically independent experiments. B.C., benzethonium chloride. **D** IB detection of FGL1 expression in indicated cells. **E** Representative H&E staining of metastatic livers from C57BL/6 J mice intraportally injected with MC38 cells followed by treatment with PBS or benzethonium chloride as indicated. Scale bar for bright-field images, 1 cm; scale bar for H&E images, 2.5 mm. **F** Quantification of liver weight and the number of liver metastases of the indicated groups of mice (*n* = 5 mice per group). **G** Representative H&E staining of metastatic livers from C57BL/6 J mice intraportally injected with MC38 cells followed by treatment as indicated. Scale bar for bright-field images, 1 cm; scale bar for H&E images, 2.5 mm.

**H** Quantification of liver weight and the number of liver metastases of the indicated groups of mice (*n* = 5 mice per group). **I–K** Flow cytometric analysis of the number of IFN-γ⁺ CD8⁺ (**I**), IFN-γ⁺ CD4⁺ (**K**), Ki67⁺ CD8⁺ and Ki67⁺ CD4⁺ T cells (**J**) in liver metastases in indicated groups of mice (*n* = 5 mice per group). **L** Survival curve analysis of mice in the indicated groups (Mean survival times of IgG+PBS, αPD-1 + PBS, IgG+B.C. and αPD-1 + B.C. mice were 26.5, 27, 33.9 and 43.5 days, respectively) (*n* = 8 mice per group). **M** Graphical abstract describing how TAMs upregulate FGL1 levels in tumor cells and mediate immune evasion as well as how benzethonium chloride reduces FGL1 in the liver microenvironment to promote antitumor immunity. The data are presented as the mean ± SD. z′ in A was calculated as previously described[40]. IB experiments in (**D**) were repeated three times, the data are representative of three biologically independent experiments. The data in (**L**) were determined by Kaplan–Meier analysis with the log-rank test. *P* values were determined by one-way ANOVA (**C**, **F**, **H**–**K**).

individual compound using the delta-FGL1 ratio with the standard deviation of the delta-FGL1 ratios ($z\ \text{Score} = \frac{Value - Mean}{SD}$) from the parent drug plate (16 drug plates total).

## Statistics and reproducibility

Student's *t* test was used for comparison between two groups to determine significant differences. Matched groups (three or more) were compared using one-way ANOVA or two-way ANOVA. The results are presented as the mean ± SD. All experiments were carried out at least three times as independent experiments with similar results, and representative images are shown. In the correlation analysis between two continuous variables, r represents the Pearson's correlation coefficients, and *p* values were calculated by Pearson's correlation test. Survival curves were plotted using the Kaplan–Meier method and compared by log-rank test. Statistical analyses were performed with GraphPad version 9.0. All statistical tests were two-sided.

## Reporting summary

Further information on research design is available in the Nature Portfolio Reporting Summary linked to this article.

## Data availability

This study analyzes existing, publicly available data. These accession numbers are available from GEO under following accession codes: GSE164522[26] and GSE31477[36]. Cohorts for estimating the association between gene expression and prognosis after immunotherapy are sourced from previously published reports[22–25]. Correlation between p65 and OTUD1 expression in CRC and GC were performed using data collected from TCGA (https://www.cancer.gov/about-nci/organization/ccg/research/structural-genomics/tcga) and CCLE (https://depmap.org/portal/download/all/). The mass spectrometry proteomics data have been deposited to the ProteomeXchange Consortium via the iProX partner repository with the dataset identifier PXD045948. The remaining data are available within the article, Supplementary information and Source data file. Source data are provided with this paper.

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

## Acknowledgements

This project was supported by the National Key R&D Program of China (2022YFA1105300), the National Natural Science Foundation of China (82341010, 82203732, 82273241, 82022052), Guangdong Basic and Applied Basic Research Foundation (2023B1515040030) and the Post-doctoral Science Foundation of China (2022M711336).

## Author contributions

H.-Q.J. and R.-H.X. designed the study. J.-J.L., J.-H.W., T.T., J.L., Y.-Q.Z., H.-Y.M., H.S., Y.-X.C., Q.-N.W. and Y.H. collected the data. H.-Q.J., J.-J.L., J.-H.W., T.T., and J.L. analyzed and interpreted the data. J.-J.L., Y.-Q.Z., and Y.-X.C. performed the statistical analysis. H.-Q.J., J.-J.L., T.T., J.L. and J.-H.W. wrote the manuscript. Z.-X.L., K.L., Y.-Q.P., Z.-L.Z. and W.Y. contributed to discussion and data interpretation. H.-Q.J., R.-H.X., K.L., J.-J.L., J.-H.W., T.T. and J.L. revised the manuscript. All authors reviewed the manuscript and approved the final version.

## Competing interests

The authors declare no competing interests.
