## [Peer Review File · Nature Communications]

The Liver Microenvironment Orchestrates FGL1-mediated Immune Escape and Progression of Metastatic Colorectal CancerREVIEWER COMMENTS

Reviewer #1 (Remarks to the Author):

In this manuscript, Li et al. describe a new crosstalk between the tumor microenvironment and metastatic colorectal cancer cells which may mediate colonization, therapy resistance and immune evasion through the secretion of FGL1, a LAG3 ligand produced by both cancer cells and hepatocytes.

As the authors stated in the introduction, metastatic colorectal cancer is a major cause of death for patients and new treatments are needed as they remain unresponsive to current therapeutic schemes. Although immune checkpoint blockade has been useful for a variety of tumors in different stages, others show resistance to these new treatments mainly due to the co-expression of other proteins which exert immune evasive traits to the tumors and metastases. The authors present a complex signalling pathway which involves tumor-associated macrophages in cancer cell and hepatocyte-overexpression of FGL1, which generates an immune suppressive phenotype in liver metastases. They present numerous experiments to dissect the complicated network of signals they want to prove, giving some insights into how metastases evade the immune systems and proposing novel compounds which can target these metastatic cells through FGL1 inhibition in both tumor cells and hepatocytes. Although they offer extensive evidence supporting their hypothesis, a limited number of additional experiments or analyses will strengthen the authors' central claims.

Specific Major comments:

1. The authors start their experimental design using a highly metastatic colorectal cancer orthotopic model with MC38 cells to validate FGL1 overexpression in metastases-derived cells (Figure 1A,B). Although MC38 are known for being highly metastatic, a duplicate experiment with another commercial or GEMM-derived cell line would strengthen their results. Also, evidence of the primary tumor and metastases formation would be helpful to fully understand the model (e.g., pictures of primary tumors and metastases, histological analyses and immunofluorescence pictures of the MC38-GFP cells in the livers).
2. Concerning the metastatic model via intraportal injection, as the authors started with a highly metastatic orthotopic model, a comparison justifying the selection of another model would be interesting as it is not very different in terms of time (24-26 days VS. 30 days). Also, the use of the orthotopic model in subsequent experiments would be of great interest as the effect of the different therapies could be checked, not only at the level of metastases, but also at the levels of primary tumor and dissemination.
3. To further validate FGL1 downregulation at the hepatocyte level, a couple of more validations are needed. First, a validation that the adenovirus targeting FGL1 do not affect tumor cells is required. Second, another validation needed would be a histological verification of FGL1 knockdown in these AAV-treated mice. Finally, as the authors goal is to validate that FGL1 inhibition reduces already established metastases, they should justify why they treat with the AAV at day two after tumor cell injection. Maybe it would be more interesting to perform this experiment once the metastases are established in the animal. The current experiment could indicate that FGL1 knockdown disrupts cancer cell seeding, instead of metastasis suppression as the authors suggested.
4. For some experiments (e.g., Figure 3D, Figure 7F) the authors employed melanoma cells or show data from other tumors types. As the main topic of this article is colorectal cancer, I

suggest that these additional experiments/data be moved in their entirety to a separate figure or supplemental figure at the end.

5. The experiment from Figure 6H-K supports that the inhibition of different immune checkpoints simultaneously reduce metastatic seeding/size (this needs to be clarified as stated at paragraph 3). Although these data are interesting, they show a big discrepancy regarding the main hypothesis of the article (i.e., current immunotherapies are not effective in CRC metastases) and are not connected with FGL1 inhibition. A combination of these immune checkpoint inhibitors plus FGL1 inhibition (via sh or B.C.) would be very clarifying as CSFR1 inhibition has been omitted from the experiments in Figure 7E-F. If aPD1+aCSFR1+shFGL1/B.C. result in better metastases reduction, it would be a great result to show and will justify the Figure 6H-K experiment. Also, performing this experiment in the orthotopic model would show effects in the primary tumor and in metastatic dissemination from the primary tumor.

6. As FGL1 is a LAG3 ligand, it would be interesting to see why FGL1 inhibition is more interesting as a therapeutical approach compared with anti-LAG3 therapies, which are currently in the clinic.

Specific Minor comments:

1. The crystal violet picture from Figure 4N could be improved to see colony differences.

Reviewer #2 (Remarks to the Author):

The aim of the work described by Li et al was to identify mechanisms that limit the efficacy of immunotherapy in patients with hepatic metastases. They identified fibrinogen-like protein 1 (FGL1), a protein secreted by hepatocytes and cancer cells, as a potential mediator of immunosuppression in the tumor microenvironment of colorectal carcinoma (CRC) liver metastases (CRCLM), using a mouse CRC model and clinical specimens. They show evidence that FGL1 production in cancer cells is regulated by $TNF\alpha$ and $IL-1\beta$ - cytokines produced by tumor-associated macrophages (TAM) and that this is mediated by transcriptional upregulation of the deubiquitinase OTUD1 that stabilizes FGL1 levels. Finally they show that Benzethonium chloride, an antiseptic, identified as an inhibitor of FGL1 secretion by screening a library of 1430 FDA approved compounds, can potentiate the effect of immunotherapy in mouse models of colon carcinoma (MC38) and melanoma (B16F10) liver metastasis.

This is a data-intensive manuscript and the authors should be commended for the thorough and comprehensive work. However there are a few fundamental issues that need to be addressed

1. The mechanism of action of FGL1 in the context of liver metastasis and in the models investigated has not been determined and remains unclear. In the models analyzed, do exhausted liver-recruited T cells express LAG3 and if yes, does FGL1 bind and block this immune checkpoint molecule? The authors speculate that it does, but the data to confirm this mode of action are lacking.

2. FGL1 is produced by hepatocytes and may also be produced by stromal cells (as suggested by experiments performed on MEF). The authors should identify the cellular sources of hepatic FGL1 and determine whether they are all regulated equally by macrophage-derived $TNF\alpha$. The relative contributions of FGL1 derived from these cellular

sources to immunosuppression is not clear.

3. TAM are a heterogeneous group of macrophages (e.g. resident Kupffer cells, recruited monocyte/macrophages, pro-inflammatory, pro-tumorigenic) with potentially opposing roles in liver metastasis progression. A better characterization of the TAM used for co-culture and subsequent experiments should be provided.

Specific comments.

Fig 1.

Fig 1D- The marked reduction in liver metastases in FGL1 KO is surprising in view of continued production of this protein by hepatocytes (as also implied by the data shown in F). This raises the possibility of additional, tumor-intrinsic functions of FGL1 (see Qian W et al, 2021 for review). This has not been discussed.

Fig 1F – The authors state that AAVs target hepatocytes specifically. What is the evidence? Also in D and F - are the H&E stained sections representative of the metastatic load for all groups?

Fig 1K- A more complete survival curve (beyond day 40) would be instructive. Do all mice eventually die of LM and if yes, what is the mean survival time for each group? This applies to other survival curves in this submission.

Fig 3:

TAMs should be phenotyped and better characterized using available markers for M1 , M2 recruited and resident macrophages.

The quality of some of the Immunoblots needs to be improved and quantification (fold change), the number of experimental replicates and p values added.

Fig 6

Fig 6B- What was the rationale for starting TNF α injections on day 9? Was this empirically determined? The authors should explain which stage in metastatic expansion and immune cell recruitment/activation this time interval corresponds to.

Fig 6 E-G: TNF α is a pleiotropic factor with multiple effects on both cancer cells and the TME, including T cell activation and Treg accumulation and survival (see for example Mehta MK, 2018). The Suggestion that its effects are mediated exclusively (or primarily) through FGL1 regulation is simplistic. For example, did the authors analyze the effects of TNF α injections on Treg, their frequency in the liver and their activity? Can Treg accumulation explain the reduction in INF γ + CD4+ and CD8+ T cells? Can TNF α at the administered dose induce apoptosis in cells of the TME?

Fig 7.

The data would be strengthened by addition of a non-active analogue of Benzethonium chloride as a control. Also, the authors should clarify how tumor cell viability was affected by this compound and whether in vivo it can affect macrophages directly. Liver images in E and F are of poor quality.

Minor points

Manuscript should be checked for spelling errors.

Examples

Line 203 and others- should be metastases

Line 228- denominator

Reviewer #3 (Remarks to the Author):

Li et al. study provides evidence on the role of FGL1 in promoting CRC metastatic progression through an immunosuppression in the liver microenvironment, by using several in vivo experimental mice models. Thus, the authors propose TAM-OTUD1-FGL1 axis as a novel therapeutic potential target for liver metastatic cancer immunotherapy. Overall, the manuscript is well-written and organized with a huge amount of data supporting the conclusions.

The following minor comments need to be further clarified:

- The authors use murine CRC cells MC38 in immunocompetent mouse models in order to demonstrate the role of FGL1 in inducing an immunosuppression in liver murine microenvironment. In vitro the authors use HT29, a human CRC cell line, with less striking results. The paper could benefit from the use of human colorectal cancer cells in vivo by orthotopic co-injection with human macrophages in order to recapitulate similar findings with the genetic inhibition of FGL1.
- The authors should better explain in details the Rag1^{-/-} mice models utilized in both Introduction and Result sections.
- The authors should improve the quality of Figure 3 panels A-B-C.
- The authors should change the title of the Results paragraph entitled “The functional roles and clinical significance of the TAM-OTUD1-FGL1 axis in CRC” better explaining the findings discussed in this paragraph.
- The authors should comment Figure 7K in the Results section not at the end of Discussion.

Manuscript ID: # NCOMMS-23-05883A

Title: The Liver Microenvironment Orchestrates FGL1-mediated Immune Escape and Progression of Metastatic Colorectal Cancer

Author Responses to Initial Comments:

Detailed point-by-point responses to the reviewers' comments:

Note to reviewers: We would like to express our sincere gratitude to the reviewers for their valuable time and constructive feedback on our manuscript. Their insightful comments have significantly contributed to enhancing the quality and scientific rigor of our work. In response to the reviewers' feedback, we have diligently revised the manuscript and incorporated new data to address all their concerns. Below are our point-by-point responses to each of the reviewers' comments. Only new data have been presented in the response letter. Please refer to the corresponding figures and text in our revised manuscript when necessary. All changes in our revised manuscript have been marked in colored text.

Reviewer #1:

In this manuscript, Li et al. describe a new crosstalk between the tumor microenvironment and metastatic colorectal cancer cells which may mediate colonization, therapy resistance and immune evasion through the secretion of FGL1, a LAG3 ligand produced by both cancer cells and hepatocytes.

As the authors stated in the introduction, metastatic colorectal cancer is a major cause of death for patients and new treatments are needed as they remain unresponsive to current therapeutic schemes. Although immune checkpoint blockade has been useful for a variety of tumors in different stages, others show resistance to these new treatments mainly due to the co-expression of other proteins which exert immune evasive traits to the tumors and metastases. The authors present a complex signalling pathway which involves tumor-associated macrophages in cancer cell and hepatocyte-

overexpression of FGL1, which generates an immune suppressive phenotype in liver metastases. They present numerous experiments to dissect the complicated network of signals they want to prove, giving some insights into how metastases evade the immune systems and proposing novel compounds which can target these metastatic cells through FGL1 inhibition in both tumor cells and hepatocytes. Although they offer extensive evidence supporting their hypothesis, a limited number of additional experiments or analyses will strengthen the authors' central claims.

Responses: We thank this reviewer for the thoughtful review of our manuscript, the positive and insightful comments, and constructive suggestions below to strengthen our hypothesis. And we hope that our revision has largely addressed the critiques from this reviewer.

Specific Major comments:

1. The authors start their experimental design using a highly metastatic colorectal cancer orthotopic model with MC38 cells to validate FGL1 overexpression in metastases-derived cells (Figure 1A, B). Although MC38 are known for being highly metastatic, a duplicate experiment with another commercial or GEMM-derived cell line would strengthen their results. Also, evidence of the primary tumor and metastases formation would be helpful to fully understand the model (e.g., pictures of primary tumors and metastases, histological analyses and immunofluorescence pictures of the MC38-GFP cells in the livers).

Responses: We thank the reviewer for providing this valuable suggestion. As suggested, we successfully established another liver metastasis mouse model of murine CRC by microsurgical orthotopic implantation of CT26-GFP cells in the cecum termini of BABL/c mice following the previous experimental procedure (**Fig 1A**). The results obtained from this model corroborate that FGL1 protein expression is significantly elevated in liver metastases-derived cells, while not observed in hepatocytes (**new Fig S1B**). And the evidence of primary tumor and liver metastases have been added (**new Fig S1A**). We have also added a brief description in revised manuscript (**Page 7 line**

123-124).

Figure legend:

(S1B) Immunoblotting (IB) detection of FGL1 in CT26 cells obtained from primary tumor tissues and liver metastatic tumor tissues, as well as in hepatocytes from control and metastatic livers. Right panel, quantitative estimates of FGL1 levels based on IB analyses.

(S1A) Representative bright-field images and H&E staining of the primary tumor and metastatic liver from C57BL/6J or BALB/c mice orthotopically implanted with MC38 cells or CT26 cells. Scale bar for bright-field images, 1 cm; scale bar for H&E images, 100 μ m.

The data were determined by one-way ANOVA. n.s., not significant. $**P < 0.01$.

2. Concerning the metastatic model via intraportal injection, as the authors started with

a highly metastatic orthotopic model, a comparison justifying the selection of another model would be interesting as it is not very different in terms of time (24-26 days VS. 30 days). Also, the use of the orthotopic model in subsequent experiments would be of great interest as the effect of the different therapies could be checked, not only at the level of metastases, but also at the levels of primary tumor and dissemination.

Responses: This reviewer asked a very insightful question. In our preliminary experiment, we observed that one hundred percent (200/200) of mice injected with MC38 or CT26 cells developed orthotopic tumors. However, the incidence of liver metastasis was relatively low, with only 5.5% (11/200) and 7.5% (15/200) for MC38 and CT26 cells, respectively, which is consistent with previous reports (Morimoto-Tomita et al., 2005; Zhang et al., 2013). Given the low incidence of liver metastasis in the orthotopic mouse model of CRC, we opted for a portal vein injection mouse model to investigate the role and regulatory mechanism of FGL1 in CRC liver metastasis (Thalheimer et al., 2009). We have added a brief explanation in revised manuscript (**Page 7 line 130-131**). We hope that this interpretation satisfies the reviewer's query and clarifies our experimental design.

3. To further validate FGL1 downregulation at the hepatocyte level, a couple of more validations are needed. First, a validation that the adenovirus targeting FGL1 do not affect tumor cells is required. Second, another validation needed would be a histological verification of FGL1 knockdown in these AAV-treated mice. Finally, as the authors goal is to validate that FGL1 inhibition reduces already established metastases, they should justify why they treat with the AAV at day two after tumor cell injection. Maybe it would be more interesting to perform this experiment once the metastases are established in the animal. The current experiment could indicate that FGL1 knockdown disrupts cancer cell seeding, instead of metastasis suppression as the authors suggested.

Responses: We thank the reviewer for this outstanding question and valuable suggestions. To downregulate FGL1 expression specifically in hepatocytes, we constructed shRNA targeting FGL1 using the pAAV-TBG-MCS carrier, which is controlled by the hepatocyte-specific thyroxine-binding globulin (TBG) promoter, as

previously reported (Liang et al., 2022). As suggested, we performed IHC and immunoblotting experiments, the results showed that adeno-associated virus (AAV) treatment effectively downregulates FGL1 expression in hepatocytes, while not affecting tumor cells (**new Fig S1G**). These data indicate that this AAV can specifically downregulate FGL1 expression in hepatocytes.

Furthermore, because the preliminary test and previous studies show that the knockdown effect of AAV takes approximately two weeks to achieve (He et al., 2022; Liang et al., 2022), and metastases were already established at this time point, we decided to treat mice with AAV at day 2 after tumor cell injection. Based on the above evidence, we speculate that FGL1 inhibition can effectively reduce already established metastases. We have added a brief description in the methods section of our revised manuscript (**Page 31 line 641-644**). We hope that the reviewer will concur with our interpretation.

Figure legend:

(**S1G**) IB detection and IHC staining of FGL1 in hepatocytes or MC38 cells from mice treated with AAV-shNC or AAV-shFGL1. Scale bar, 100 μm.

4. For some experiments (e.g., Figure 3D, Figure 7F) the authors employed melanoma cells or show data from other tumors types. As the main topic of this article is colorectal cancer, I suggest that these additional experiments/data be moved in their entirety to a

separate figure or supplemental figure at the end.

Responses: We thank the reviewer for pointing out this issue. As suggested, we have moved the data from other tumor types to the supplemental figure section in revised manuscript (**new Fig S6**).

5. The experiment from Figure 6H-K supports that the inhibition of different immune checkpoints simultaneously reduce metastatic seeding/size (this needs to be clarified as stated at paragraph 3). Although these data are interesting, they show a big discrepancy regarding the main hypothesis of the article (i.e., current immunotherapies are not effective in CRC metastases) and are not connected with FGL1 inhibition. A combination of these immune checkpoint inhibitors plus FGL1 inhibition (via sh or B.C.) would be very clarifying as CSFR1 inhibition has been omitted from the experiments in Figure 7E-F. If aPD1+aCSFR1+shFGL1/B.C. result in better metastases reduction, it would be a great result to show and will justify the Figure 6H-K experiment. Also, performing this experiment in the orthotopic model would show effects in the primary tumor and in metastatic dissemination from the primary tumor.

Responses: We thank this reviewer for raising this crucial question, and we sincerely apologize for any confusion caused by the unclear description of our experiment. As depicted in **Fig 3** of our manuscript, TAM plays a significant role in promoting the stabilization of FGL1 within the liver microenvironment. To evaluate the combination of FGL1 inhibition and anti-PD-1 therapy, we employed an anti-CSF1R antibody to deplete TAM, following the approach used in a previous report (Zhang et al., 2020). It is important to note that the anti-CSF1R antibody was not utilized as an immune checkpoint inhibitor in this study. In addition, regarding the low incidence of liver metastasis observed in the orthotopic mouse model of CRC as mentioned above (**Question 2**), this experiment was performed using the portal vein injection mouse model. We have provided clearer descriptions of these experimental details in our revised manuscript (**Page 17 line 339-340**).

6. As FGL1 is a LAG3 ligand, it would be interesting to see why FGL1 inhibition is

more interesting as a therapeutical approach compared with anti-LAG3 therapies, which are currently in the clinic.

Responses: This reviewer raised an outstanding question. LAG-3 (CD223) is an inhibitory receptor primarily expressed on activated T cells, and its binding with ligands leads to T cell dysfunction and potentially inhibits antitumor immunity (Maruhashi et al., 2020; Qian et al., 2021). Initially identified in 1990, LAG-3 was recognized as a novel lymphocyte activation gene closely related to CD4 (Triebel et al., 1990). Over the course of more than three decades of investigations, multiple clinical trials are now underway to test the efficacy of LAG-3-targeted therapy in various cancers, either as monotherapy or in combination, primarily with anti-PD-1/PD-L1 antibodies (Maruhashi et al., 2020; Tawbi et al., 2022). In 2019, FGL1 was identified as the major ligand of LAG-3 (Wang et al., 2019), further emphasizing the importance of preclinical studies and clinical trials related to FGL1. We believe that FGL1 holds great potential as an immune checkpoint target in clinical practice in the near future. This discussion has been elaborated upon in our revised manuscript (**Page 21 line 429-431**). We hope that this interpretation will address the reviewer's query.

Specific Minor comments:

1. The crystal violet picture from Figure 4N could be improved to see colony differences.

Responses: We thank the reviewer for pointing this out. We have duplicated the experiment and replaced the crystal violet picture of Figure 4N (**new Fig 4N**).

Figure legend:

(4N) Representative images and quantification of the indicated cells cocultured with activated OT-1 T cells for 8 hr, followed by crystal violet staining. The ratio of MC38-OVA cells to OT-1 cells was 5:1. Scale bar, 10 μ m.

The data were determined by one-way ANOVA. * $P < 0.05$; ** $P < 0.01$.

Reviewer #2:

The aim of the work described by Li et al was to identify mechanisms that limit the efficacy of immunotherapy in patients with hepatic metastases. They identified fibrinogen-like protein 1 (FGL1), a protein secreted by hepatocytes and cancer cells, as a potential mediator of immunosuppression in the tumor microenvironment of colorectal carcinoma (CRC) liver metastases (CRCLM), using a mouse CRC model and clinical specimens. They show evidence that FGL1 production in cancer cells is regulated by TNF α and IL-1 β - cytokines produced by tumor-associated macrophages (TAM) and that this is mediated by transcriptional upregulation of the deubiquitinase OTUD1 that stabilizes FGL1 levels. Finally they show that Benzethonium chloride, an antiseptic, identified as an inhibitor of FGL1 secretion by screening a library of 1430 FDA approved compounds, can potentiate the effect of immunotherapy in mouse models of colon carcinoma (MC38) and melanoma (B16F10) liver metastasis.

This is a data-intensive manuscript and the authors should be commended for the thorough and comprehensive work. However there are a few fundamental issues that need to be addressed.

Responses: We thank the reviewer for appreciating our study and providing valuable comments to enhance the quality of our manuscript. We hope that our revision has largely addressed the critiques raised by this reviewer.

1. The mechanism of action of FGL1 in the context of liver metastasis and in the models investigated has not been determined and remains unclear. In the models analyzed, do exhausted liver-recruited T cells express LAG3 and if yes, does FGL1 bind and block this immune checkpoint molecule? The authors speculate that it does, but the data to confirm this mode of action are lacking.

Responses: This reviewer posed an insightful question. To our knowledge, FGL1 has been reported to have the capacity to bind LAG-3 and is a major ligand of LAG-3, as previously reported in the literature (Wang et al., 2019). However, we acknowledged that our initial manuscript lacked sufficient data to elucidate this interaction. To address

this question, we performed flow cytometry experiments, revealing the expression of LAG-3 on exhausted liver-recruited T cells (**new Fig S1K**), which is consistent with previous reports (Maruhashi et al., 2020; Triebel et al., 1990).

Furthermore, to investigate the binding between FGL1 and LAG-3, we performed an enzyme-linked immunosorbent assay (ELISA) and confirmed that FGL1 does bind to LAG-3 *in vitro* (**new Fig S1J**). Interestingly, our experimental results demonstrated that the administration of FGL1 significantly reduced IFN- γ production by activated splenic T cells, which are known to highly express LAG-3 (Maruhashi et al., 2020; Triebel et al., 1990). Additionally, we found that blocking LAG-3 effectively rescued the reduced levels of IFN- γ production (**new Fig S1L**). These findings strongly suggest that FGL1 binds to and blocks LAG-3 on activated T cells. In response to the suggestion provided by the reviewer, we have included related description in revised manuscript (**Page 9 line 157-164**).

Figure legend:

(**S1K**) Flow cytometric analysis of LAG-3 expression on PD1⁺ TIM3⁺ CD8⁺ T cells derived from liver metastases. The Fluorescence Minus One (FMO) control was used

to determine LAG-3 positivity.

(S1J) Binding curve of recombinant hFGL1 on recombinant hLAG-3.

(S1L) ELISA detection of IFN- γ secreted from splenic T cells treated as indicated.

The data were determined by one-way ANOVA. * $P < 0.05$; ** $P < 0.01$.

2. FGL1 is produced by hepatocytes and may also be produced by stromal cells (as suggested by experiments performed on MEF). The authors should identify the cellular sources of hepatic FGL1 and determine whether they are all regulated equally by macrophage-derived TNF α . The relative contributions of FGL1 derived from these cellular sources to immunosuppression is not clear.

Responses: We thank the reviewer for this outstanding question. To our knowledge, FGL1 is predominantly expressed in the liver under normal physiological conditions (Liu et al., 2022b; Qian et al., 2021). To address this query effectively, we conducted a thorough analysis comparing the expression of FGL1 in various cell types within the liver microenvironment, using published single-cell RNA-sequencing data. The results showed that FGL1 specifically exhibits high expression levels in hepatocytes, while its expression in stromal cells is relatively low (Lu et al., 2022; Werba et al., 2023) (**Figure below A**), indicating that the main sources of hepatic FGL1 in liver metastases are hepatocytes and cancer cells. This discrepancy could potentially be attributed to the fact that the mouse embryonic fibroblast (MEF) cells utilized as a tool in this study is an immortalized cell line, it might not fully represent the characteristics of primary stromal cells present in the liver microenvironment.

Furthermore, we investigated the impact of TNF α treatment on primary hepatocytes and our findings indicate that TNF α treatment does not significantly affect primary hepatocytes (**Figure below B**). This may be attributed to the distinct signal transduction mechanisms between tumor cells and normal cells. Thus, as is shown in **new Fig 1D-H**, our data suggest that hepatic FGL1 in liver metastases originates from both hepatocytes and cancer cells, both of which contribute to immunosuppression within the tumor microenvironment. We have included related description in our revised

Figure legend:

(A) FGL1 expression in different types of cells based on scRNA-seq data.

(B) IB detection of FGL1 expression in primary hepatocytes sorted from different mice then stimulated with TNF α (5 or 10 ng/mL) for 16 hr.

3. TAM are a heterogenous group of macrophages (e.g. resident Kupffer cells, recruited monocyte/macrophages, pro-inflammatory, pro-tumorigenic) with potentially opposing roles in liver metastasis progression. A better characterization of the TAM used for co-culture and subsequent experiments should be provided.

Responses: We thank the reviewer for this valuable suggestion. In response, we performed qPCR and flow cytometry analyses to investigate the markers of TAMs in the specified tissues. Our results showed that TAMs from liver metastases predominantly expressed CD163, Arg-1, and CD206, while exhibiting minimal expression of iNOS and IL-12 (new Fig S3A and S3B). This pattern of marker expression suggests that the TAMs in liver metastases tend to display an anti-inflammatory or M2 subtype, in line with the prevailing reports indicating that TAMs in most tumors are predominantly of the M2 subtype (Liu et al., 2022a; Mantovani et al., 2002). We have included these data into our revised figures and added related description in revised manuscript (Page 11 line 213-215).

Figure legend:

(S3A) qPCR and flow cytometric analysis of the markers of the TAMs from different mouse liver metastases and control cases.

(S3B) qPCR and flow cytometric analysis of the markers of the TAMs from different CRLM patients and control cases.

The data were determined by one-way ANOVA and Student's *t* test. n.s., not significant.

P* < 0.05; *P* < 0.01.

Specific comments:

1. Fig 1.

Fig 1D- The marked reduction in liver metastases in FGL1 KO is surprising in view of continued production of this protein by hepatocytes (as also implied by the data shown in F). This raises the possibility of additional, tumor-intrinsic functions of FGL1 (see Qian W et al, 2021 for review). This has not been discussed.

Responses: We appreciate the insightful question raised by the reviewer. As demonstrated in **new Fig 1D-F**, the loss of hepatic FGL1 from either tumor cells or hepatocytes leads to a marked suppression of the metastatic tumor progression *in vivo*. However, we observed that the downregulation of FGL1 specifically in hepatocytes resulted in a relatively weaker effect on the reduction of liver metastases (**new Fig 1D-F**). This discrepancy may be attributed to the time required to achieve effective knockdown, and the preliminary test and previous studies show that the knockdown effect of AAV takes approximately two weeks to achieve (He et al., 2022; Liang et al., 2022). To address this point, we have included a brief discussion in our revised manuscript (**Page 8 line 147-149**).

2. Fig 1F – The authors state that AAVs target hepatocytes specifically. What is the evidence?

Responses: We appreciate the significant question raised by the reviewer. To effectively downregulate FGL1 expression in hepatocytes, we constructed short hairpin RNA (shRNA) targeting FGL1 using the pAAV-TBG-MCS carrier, which is under the control of hepatocyte-specific thyroxine-binding globulin (TBG) promoter (Liang et al., 2022). Moreover, we performed IHC and immunoblotting experiments to evaluate the efficacy of our treatment strategy. The results indicate that treatment with adeno-associated virus (AAV) led to a notable downregulation of FGL1 expression in hepatocytes, while showing no significant impact on FGL1 expression in tumor cells (**new Fig S1G**), suggesting that the AAV-mediated approach efficiently and specifically targets FGL1 expression in hepatocytes. We have added related description in revised manuscript (**Page 8 line 144-147**).

Figure legend:

(S1G) IB detection and IHC staining of FGL1 in hepatocytes or MC38 cells from mice treated with AAV-shNC or AAV-shFGL1. Scale bar, 100 μ m.

3. Also in D and F - are the H&E stained sections representative of the metastatic load for all groups?

Responses: We thank this reviewer for asking this important question. As the reviewer pointed out, the H&E stained sections provided representative snapshots of the metastatic load in all experimental groups. Additionally, we also showed quantitative data of liver weight and the number of liver metastases for each group in **new Fig 1G-H**. We hope that this reviewer will be satisfied with our interpretation.

4. Fig 1K- A more complete survival curve (beyond day 40) would be instructive. Do all mice eventually die of LM and if yes, what is the mean survival time for each group? This applies to other survival curves in this submission.

Responses: This reviewer asked a very insightful question. As suggested, we have repeated the experiments and replaced the relevant data in our revised manuscript (**new Fig 1K, 6D, 7L and S6K**). All mice with intraportal transplantation eventually die of liver metastasis. And we have added a brief description of the mean survival time for each group in the respective **figure legends** of our revised manuscript.

Figure legend:

(1K) Survival curve analysis of mice with FGL1 knockdown in MC38 cells and hepatocytes (Mean survival times of shNC, shFGL1 #1, shFGL1 #2, AAV-shNC, AAV-shFGL1 #1 and AAV-shFGL1 #2 mice were 27.5, 41.6, 40.8, 24.5, 35 and 35.8 days, respectively).

(6D) Survival curve analysis of mice in the indicated groups (Mean survival times of Ctrl+PBS, Ctrl+TNF α , shOTUD1+TNF α , shFGL1+TNF α and shFGL1+rFGL1+TNF α mice were 25.8, 19.1, 27.6, 32.5 and 21.5 days, respectively).

(7L) Survival curve analysis of mice implanted with MC38 cells via portal vein injection followed by treatment as indicated (Mean survival times of IgG+PBS, α PD-1+PBS, IgG+B.C. and α PD-1+B.C. mice were 26.5, 27, 33.9 and 43.5 days, respectively).

(S6K) Survival curve analysis of mice implanted with B16F10 cells via portal vein injection followed by treatment as indicated (Mean survival times of IgG+PBS, α PD-1+PBS, IgG+B.C. and α PD-1+B.C. mice were 21.5, 22.9, 32 and 41 days, respectively).

The data were determined by Kaplan-Meier analysis with the log-rank test. n.s., not

significant. ** $P < 0.01$.

5. Fig 3:

TAMs should be phenotyped and better characterized using available markers for M1, M2 recruited and resident macrophages.

Responses: We appreciate the reviewer for the comment. As mentioned above (**Question 3**), our results indicate that TAMs from liver metastases show pro-tumorigenic or M2 subtype evidenced by elevated expression of CD163, Arg-1 and CD206 (**new Fig S3A and S3B**), which is consistent with previous reports (Liu et al., 2022a; Mantovani et al., 2002). We have added a brief description in revised manuscript (**Page 11 line 213-215**).

6. The quality of some of the Immunoblots needs to be improved and quantification (fold change), the number of experimental replicates and p values added.

Responses: We thank the reviewer for pointing this out. As suggested, we have repeated the experiments to improve the quality of certain immunoblots. And we have double checked and added the number of experimental replicates and p values in our revised figure legends.

7. Fig 6

Fig 6B- What was the rationale for starting TNF α injections on day 9? Was this empirically determined? The authors should explain which stage in metastatic expansion and immune cell recruitment/activation this time interval corresponds to.

Responses: We appreciate the reviewer for raising this important question. We performed preliminary experiments to monitor the plasma FGL1 levels in mice after MC38 transplantation. The results showed a continuous increase in plasma FGL1 levels over time, and it reached a two-fold increase approximately one week after MC38 transplantation (**new Fig S5A**). Thus, we speculated that the metastases have been established at this time point, and the results confirmed the establishment of metastases (**new Fig S5B**). Concurrently, flow cytometric analysis demonstrated a significant

increase in the number of IFN- γ ⁺ CD8⁺ and IFN- γ ⁺ CD4⁺ T cells compared with normal liver (**new Fig S5C**). These findings indicate that metastases were successfully established approximately one week after MC38 transplantation and that T cells had been activated at this time point. Considering the evidence obtained from these preliminary experiments, we chose to initiate TNF α injection on day 9. We have added a brief description in our revised manuscript (**Page 16 line 321-324**).

Figure legend:

(S5A) ELISA detection of plasma FGL1 levels in mice treated as indicated (n=3 mice per group).

(S5B) Representative bright-field images and H&E staining of liver tissues from mice treated as indicated. Scale bar for bright-field images, 1 cm; scale bar for H&E images, 100 μ m.

(S5C) Flow cytometric analysis of the number of IFN- γ ⁺ CD8⁺, IFN- γ ⁺ CD4⁺ T cells in liver metastases or normal liver from mice treated as indicated.

The data were determined by Student's *t* test. **P* < 0.05.

8. Fig 6 E-G: TNF α is a pleiotropic factor with multiple effects on both cancer cells and the TME, including T cell activation and Treg accumulation and survival (see for example Mehta MK, 2018). The Suggestion that its effects are mediated exclusively (or primarily) through FGL1 regulation is simplistic. For example, did the authors analyze the effects of TNF α injections on Treg, their frequency in the liver and their activity? Can Treg accumulation explain the reduction in INF- γ ⁺ CD4⁺ and CD8⁺ T cells? Can

TNF α at the administered dose induce apoptosis in cells of the TME?

Responses: We appreciate the insightful question raised by the reviewer. As suggested, we performed flow cytometry experiments to investigate the effects of TNF α treatment on Treg cells in the context of hepatic metastases. Our findings indicate that treatment with indicated dose of TNF α slightly increased the frequency of Treg cells within the CD4⁺ T cell population, albeit not reaching statistical significance. Nevertheless, treatment with TNF α significantly increased the absolute number of Treg cells in liver metastases (**new Fig S5E**), in agreement with previous study (Lexmond et al., 2016). Further investigation revealed that the expression of CTLA-4 on Treg cells and the frequency of apoptotic Treg cells in liver metastases remained unaffected when treated with TNF α (Wing et al., 2008) (**new Fig S5F and Figure below A**), indicating that treatment with the indicated dose of TNF α did not significantly alter Treg cell activity and apoptosis in the context of hepatic metastases.

While the accumulation of Treg cells mediated by TNF α may indeed contribute to the reduction in anticancer immunity, our study primarily focuses on investigating the role of FGL1 in liver metastases. Then we performed additional experiments, wherein we duplicated the study and rescued FGL1-deleted cells with short hairpin RNA (shRNA)-resistant FGL1 (rFGL1). The results demonstrated a significant reduction in hepatic metastases upon deletion of FGL1 (**new Fig 6A and 6B**), which could be rescued by overexpressing rFGL1. These findings support the critical role of FGL1 in TNF α -mediated progression of metastatic tumors. Interestingly, the administration of TNF α promoted the progression of metastatic tumors in mice implanted with FGL1-deleted MC38 cells, suggesting that TNF α also facilitates tumor progression through alternative mechanisms, potentially involving Treg cells. In light of the reviewer's concern, we have included a brief discussion in our revised manuscript (**Page 16 line 328-330 and Page 17 line 331-334**).

Figure legend:

(S5E) Flow cytometric analysis of the frequency and the number of CD4⁺ CD25⁺ Foxp3⁺ cells within CD4⁺ T cells in liver metastases from mice treated as indicated.

(S5F) Flow cytometric analysis of CTLA-4 expression on Treg cells within liver metastases from mice treated as indicated.

(A) Flow cytometric analysis of the frequency of apoptotic Treg cells in liver metastases from mice treated as indicated.

(6A) Representative H&E staining of metastatic livers from C57BL/6J mice implanted

with the control, OTUD1 knockdown, FGL1 knockdown or FGL1 knockdown with overexpression of short hairpin RNA (RNA)-resistant FGL1 (rFGL1) MC38 cells via intraportal vein injection followed by TNF α treatment. Scale bar for bright-field images, 1 cm; scale bar for H&E images, 2.5 mm.

(6B) Quantification of liver weight and the number of liver metastases of the indicated groups of mice (n=5 mice per group).

The data were determined by Student's *t* test and one-way ANOVA. n.s., not significant. **P* < 0.05; ***P* < 0.01.

7. Fig 7.

The data would be strengthened by addition of a non-active analogue of Benzethonium chloride as a control. Also, the authors should clarify how tumor cell viability was affected by this compound and whether *in vivo* it can affect macrophages directly.

Responses: We appreciate the reviewer for pointing this out. Benzethonium chloride is a synthetic quaternary ammonium salt known for its antiseptic and anti-infective properties (De Mario et al., 2021). In this study, we revealed that benzethonium chloride inhibits FGL1 secretion, but its functional structure still needs further investigation, so we used the solvent (PBS) as a control.

As suggested by the reviewer, we performed experiment to investigate the effects of benzethonium chloride at the indicated doses on tumor cell viability. The results showed that treatment with benzethonium chloride did not affect the viability of tumor cells (**new Fig S6A**). Additionally, flow cytometry analysis demonstrated that benzethonium chloride treatment did not directly impact macrophages in the tumor microenvironment *in vivo* (**new Fig S6E**). Thus, the primary mode of action appears to be the inhibition of FGL1 secretion. We have added related description in revised manuscript (**Page 19 line 379-380 and line 387-388**).

Figure legend:

(S6A) Cell viability assay (MTS) analysis of MC38 cells and HT29 cells treated with indicated doses of benzethonium chloride for 12 hr or 24 hr.

(S6E) Flow cytometric analysis of the number of TAMs in liver metastases from mice treated as indicated.

The data were determined by one-way ANOVA (S6A) and Student's *t* test (S6E). n.s., not significant.

8. Liver images in E and F are of poor quality.

Responses: We thank the reviewer for pointing out this issue, and we have improved the quality of the liver images in our revised manuscript (new Fig 7G and S6G).

Minor points:

1. Manuscript should be checked for spelling errors.

Examples

Line 203 and others- should be metastases

Line 228- denominator

Responses: We sincerely apologize for the mistakes, and we have revised the spelling errors.

Reviewer #3:

Li et al. study provides evidence on the role of FGL1 in promoting CRC metastatic progression through an immunosuppression in the liver microenvironment, by using several in vivo experimental mice models. Thus, the authors propose TAM-OTUD1-FGL1 axis as a novel therapeutic potential target for liver metastatic cancer immunotherapy. Overall, the manuscript is well-written and organized with a huge amount of data supporting the conclusions.

Responses: We appreciated the positive and insightful comments, as well as the valuable suggestions from this reviewer. We hope that our revision has largely addressed the critiques raised by this reviewer.

Specific Minor comments:

1. The authors use murine CRC cells MC38 in immunocompetent mouse models in order to demonstrate the role of FGL1 in inducing an immunosuppression in liver murine microenvironment. In vitro the authors use HT29, a human CRC cell line, with less striking results. The paper could benefit from the use of human colorectal cancer cells in vivo by orthotopic co-injection with human macrophages in order to recapitulate similar findings with the genetic inhibition of FGL1.

Responses: This reviewer posed a very insightful question. As we previously described (Li et al., 2022), we generated humanized mice by transferring human peripheral blood mononuclear cells (PBMC) into NCG mice, and observed that co-injection of HT29 cells with TAMs significantly increased tumor burden in the liver using intraportal transplantation mouse model, as evidenced by the pathology analysis, liver weight and the number of liver metastases (**new Fig S5G and S5H**). These findings further demonstrated that TAMs facilitate metastatic CRC progression, corroborating the data presented in **Fig 3** and **Fig 6** of our study. In response to the reviewer's valuable suggestion, we have included these data in our revised manuscript (**Page 17 line 336-339**).

Figure legend:

(S5G) Representative bright-field images and H&E staining of liver metastases from mice treated as indicated. Scale bar for bright-field images, 1 cm; scale bar for H&E images, 1.25 mm.

(S5H) Quantification of liver weight and the number of liver metastases of indicated groups of mice (n=5 mice per group).

The data were determined by Student's *t* test. ** $P < 0.01$.

2. The authors should better explain in details the Rag1^{-/-} mice models utilized in both Introduction and Result sections.

Responses: We are thankful to the reviewer for raising this insightful question. Rag1^{-/-} mice exhibit a defect in V(D)J recombination, a process mediated by recombination activation genes (RAG-1 and RAG-2), which is crucial for the maturation of B and T cells. Consequently, Rag1^{-/-} mice have immature B and T cells (Mombaerts et al., 1992; Shinkai et al., 1992). We have included a brief description in our revised manuscript (**Page 8 line 134-135 and line 151-153**).

3. The authors should improve the quality of Figure 3 panels A-B-C.

Responses: We thank the reviewer for pointing this out. As suggested, we have optimized the clustering analysis of the scRNA-seq data and improved the quality of the images (**new Fig 3A, 3B and 3C**) presented in our revised manuscript.

4. The authors should change the title of the Results paragraph entitled “The functional

roles and clinical significance of the TAM-OTUD1-FGL1 axis in CRC” better explaining the findings discussed in this paragraph.

Responses: We thank the reviewer for pointing this out. As suggested, we have changed the title as “The TAM/TNF α -OTUD1-FGL1 axis promotes immune escape and progression of CRLM” (Page 16 line 318-319).

5. The authors should comment Figure 7K in the Results section not at the end of Discussion.

Responses: We thank the reviewer for pointing out this issue. As suggested, we have moved the description of Figure 7K (new Fig 7M) to the results section in our revised manuscript (Page 20 line 403-405).

References:

- De Mario, A., A. Tosatto, J.M. Hill, J. Kriston-Vizi, R. Ketteler, D. Vecellio Reane, G. Cortopassi, G. Szabadkai, R. Rizzuto, and C. Mammucari. 2021. Identification and functional validation of FDA-approved positive and negative modulators of the mitochondrial calcium uniporter. *Cell Rep* 35:109275.
- He, X., Z. Zhang, J. Xue, Y. Wang, S. Zhang, J. Wei, C. Zhang, J. Wang, B.A. Urip, C.C. Ngan, J. Sun, Y. Li, Z. Lu, H. Zhao, D. Pei, C.K. Li, and B. Feng. 2022. Low-dose AAV-CRISPR-mediated liver-specific knock-in restored hemostasis in neonatal hemophilia B mice with subtle antibody response. *Nature communications* 13:7275.
- Lexmond, W.S., J.A. Goettel, J.J. Lyons, J. Jacobse, M.M. Deken, M.G. Lawrence, T.H. DiMaggio, D. Kotlarz, E. Garabedian, P. Sackstein, C.C. Nelson, N. Jones, K.D. Stone, F. Candotti, E.H. Rings, A.J. Thrasher, J.D. Milner, S.B. Snapper, and E. Fiebiger. 2016. FOXP3+ Tregs require WASP to restrain Th2-mediated food allergy. *J Clin Invest* 126:4030-4044.
- Li, T., Y.T. Tan, Y.X. Chen, X.J. Zheng, W. Wang, K. Liao, H.Y. Mo, J. Lin, W. Yang, H.L. Piao, R.H. Xu, and H.Q. Ju. 2022. Methionine deficiency facilitates antitumour immunity by altering m(6)A methylation of immune checkpoint transcripts. *Gut*
- Liang, C.Q., D.C. Zhou, W.T. Peng, W.Y. Chen, H.Y. Wu, Y.M. Zhou, W.L. Gu, K.S. Park, H. Zhao, L.Q. Pi, L. Zheng, S.S. Feng, D.Q. Cai, and X.F. Qi. 2022. FoxO3 restricts liver regeneration by suppressing the proliferation of hepatocytes. *NPJ Regenerative medicine* 7:33.
- Liu, J., Z. Liu, J. Li, Z. Zeng, J. Wang, X. Luo, C. Wong, J. Zheng, H. Pu, H. Mo, H. Sheng, Q. Wu, H. Li, G. Wan, B. Li, D. Wang, R. Xu, and H.J.C.r. Ju. 2022a. The macrophage-associated lncRNA MALR facilitates ILF3 liquid-liquid phase separation to promote HIF1 α signaling in esophageal cancer.
- Liu, X.H., L.W. Qi, R.N. Alolga, and Q. Liu. 2022b. Implication of the hepatokine, fibrinogen-like protein 1 in liver diseases, metabolic disorders and cancer: The need to harness its full potential. *Int J*

Biol Sci 18:292-300.

- Lu, Y., A. Yang, C. Quan, Y. Pan, H. Zhang, Y. Li, C. Gao, H. Lu, X. Wang, P. Cao, H. Chen, S. Lu, and G. Zhou. 2022. A single-cell atlas of the multicellular ecosystem of primary and metastatic hepatocellular carcinoma. *Nature communications* 13:4594.
- Mantovani, A., S. Sozzani, M. Locati, P. Allavena, and A. Sica. 2002. Macrophage polarization: tumor-associated macrophages as a paradigm for polarized M2 mononuclear phagocytes. *Trends in immunology* 23:549-555.
- Maruhashi, T., D. Sugiura, I.M. Okazaki, and T. Okazaki. 2020. LAG-3: from molecular functions to clinical applications. *Journal for immunotherapy of cancer* 8:
- Mombaerts, P., J. Iacomini, R.S. Johnson, K. Herrup, S. Tonegawa, and V.E. Papaioannou. 1992. RAG-1-deficient mice have no mature B and T lymphocytes. *Cell* 68:869-877.
- Morimoto-Tomita, M., Y. Ohashi, A. Matsubara, M. Tsuiji, and T. Irimura. 2005. Mouse colon carcinoma cells established for high incidence of experimental hepatic metastasis exhibit accelerated and anchorage-independent growth. *Clinical & experimental metastasis* 22:513-521.
- Qian, W., M. Zhao, R. Wang, and H. Li. 2021. Fibrinogen-like protein 1 (FGL1): the next immune checkpoint target. *Journal of hematology & oncology* 14:147.
- Shinkai, Y., G. Rathbun, K.P. Lam, E.M. Oltz, V. Stewart, M. Mendelsohn, J. Charron, M. Datta, F. Young, and A.M. Stall. 1992. RAG-2-deficient mice lack mature lymphocytes owing to inability to initiate V(D)J rearrangement. *Cell* 68:855-867.
- Tawbi, H.A., D. Schadendorf, E.J. Lipson, P.A. Ascierto, L. Matamala, E. Castillo Gutierrez, P. Rutkowski, H.J. Gogas, C.D. Lao, J.J. De Menezes, S. Dalle, A. Arance, J.J. Grob, S. Srivastava, M. Abaskharoun, M. Hamilton, S. Keidel, K.L. Simonsen, A.M. Sobiesk, B. Li, F.S. Hodi, G.V. Long, and R.-. Investigators. 2022. Relatlimab and Nivolumab versus Nivolumab in Untreated Advanced Melanoma. *N Engl J Med* 386:24-34.
- Thalheimer, A., C. Otto, M. Bueter, B. Illert, S. Gattenlohner, M. Gasser, D. Meyer, M. Fein, C.T. Germer, and A.M. Waaga-Gasser. 2009. The intraportal injection model: a practical animal model for hepatic metastases and tumor cell dissemination in human colon cancer. *BMC Cancer* 9:29.
- Triebel, F., S. Jitsukawa, E. Baixeras, S. Roman-Roman, C. Genevee, E. Viegas-Pequignot, and T. Hercend. 1990. LAG-3, a novel lymphocyte activation gene closely related to CD4. *J Exp Med* 171:1393-1405.
- Wang, J., M.F. Sanmamed, I. Datar, T.T. Su, L. Ji, J. Sun, L. Chen, Y. Chen, G. Zhu, W. Yin, L. Zheng, T. Zhou, T. Badri, S. Yao, S. Zhu, A. Boto, M. Sznol, I. Melero, D.A.A. Vignali, K. Schalper, and L. Chen. 2019. Fibrinogen-like Protein 1 Is a Major Immune Inhibitory Ligand of LAG-3. *Cell* 176:334-347 e312.
- Werba, G., D. Weissinger, E.A. Kawaler, E. Zhao, D. Kalfakakou, S. Dhara, L. Wang, H.B. Lim, G. Oh, X. Jing, N. Beri, L. Khanna, T. Gonda, P. Oberstein, C. Hajdu, C. Loomis, A. Heguy, M.H. Sherman, A.W. Lund, T.H. Welling, I. Dolgalev, A. Tsirigos, and D.M. Simeone. 2023. Single-cell RNA sequencing reveals the effects of chemotherapy on human pancreatic adenocarcinoma and its tumor microenvironment. *Nature communications* 14:797.
- Wing, K., Y. Onishi, P. Prieto-Martin, T. Yamaguchi, M. Miyara, Z. Fehervari, T. Nomura, and S. Sakaguchi. 2008. CTLA-4 control over Foxp3+ regulatory T cell function. *Science* 322:271-275.
- Zhang, L., Z. Li, K.M. Skrzypczynska, Q. Fang, W. Zhang, S.A. O'Brien, Y. He, L. Wang, Q. Zhang, A. Kim, R. Gao, J. Orf, T. Wang, D. Sawant, J. Kang, D. Bhatt, D. Lu, C.M. Li, A.S. Rapaport, K. Perez, Y. Ye, S. Wang, X. Hu, X. Ren, W. Ouyang, Z. Shen, J.G. Egen, Z. Zhang, and X. Yu. 2020. Single-Cell

Analyses Inform Mechanisms of Myeloid-Targeted Therapies in Colon Cancer. *Cell* 181:442-459 e429.

Zhang, Y., C. Davis, J. Ryan, C. Janney, and M.M. Peña. 2013. Development and characterization of a reliable mouse model of colorectal cancer metastasis to the liver. *Clinical & experimental metastasis* 30:903-918.

REVIEWERS' COMMENTS

Reviewer #1 (Remarks to the Author):

The authors have adequately addressed all of our concerns, and have presented a substantial amount of additional data that has improved the manuscript.

Reviewer #2 (Remarks to the Author):

The additional data addressed many of the concerns and added mechanistic insight into the findings.

Reviewer #3 (Remarks to the Author):

The authors have responded to all the comments of this Reviewer.